# The RNA interactome of human telomerase RNA reveals a coding-independent role for a histone mRNA in telomere homeostasis

Roland Ivanyi-Nagy[1]*, Syed Moiz Ahmed[1], Sabrina Peter[1], Priya Dharshana Ramani[1], Peh Fern Ong[2], Oliver Dreesen[2], Peter Dröge[1,3]*

[1]School of Biological Sciences, Nanyang Technological University, Singapore, Singapore; [2]Cell Ageing, Skin Research Institute Singapore, Singapore, Singapore; [3]Nanyang Institute of Structural Biology, Nanyang Technological University, Singapore, Singapore

**Abstract** Telomerase RNA (TR) provides the template for DNA repeat synthesis at telomeres and is essential for genome stability in continuously dividing cells. We mapped the RNA interactome of human TR (hTR) and identified a set of non-coding and coding hTR-interacting RNAs, including the histone 1C mRNA (*HIST1H1C*). Disruption of the hTR-*HIST1H1C* RNA association resulted in markedly increased telomere elongation without affecting telomerase enzymatic activity. Conversely, over-expression of *HIST1H1C* led to telomere attrition. By using a combination of mutations to disentangle the effects of histone 1 RNA synthesis, protein expression, and hTR interaction, we show that *HIST1H1C* RNA negatively regulates telomere length independently of its protein coding potential. Taken together, our data provide important insights into a surprisingly complex hTR-RNA interaction network and define an unexpected non-coding RNA role for *HIST1H1C* in regulating telomere length homeostasis, thus offering a glimpse into the mostly uncharted, vast space of non-canonical messenger RNA functions.
DOI: https://doi.org/10.7554/eLife.40037.001

*For correspondence:
roland.ivanyi-nagy@ntu.edu.sg (RI-N);
pdroge@ntu.edu.sg (PD)

**Competing interests:** The authors declare that no competing interests exist.

## Introduction

Most human cells display progressive telomere shortening during cell divisions, ultimately resulting in replicative senescence or apoptosis (*Harley et al., 1990*; *Maciejowski and de Lange, 2017*). In the majority of cancer cells and in continuously dividing germ line cells, however, telomere erosion is mitigated by the action of telomerase – a specialized ribonucleoprotein (RNP) complex minimally composed of telomerase RNA (TR) and the telomerase reverse transcriptase (TERT) enzyme. Telomere homeostasis depends on the highly regulated co-ordination of telomerase RNP assembly, trafficking, and recruitment to telomeres during the S phase of the cell cycle (*Schmidt and Cech, 2015*).

While all TRs contain a short internal template for telomeric DNA repeat synthesis (*Greider and Blackburn, 1989*), vertebrate TRs also possess an H/ACA box small Cajal body (CB)-specific RNA (scaRNA) domain (*Jády et al., 2004*; *Mitchell et al., 1999a*) (*Figure 1A*) that associates with the canonical H/ACA scaRNA-binding proteins (*Nguyen et al., 2018*), including the pseudouridine synthase dyskerin (*Mitchell et al., 1999b*) and the CB chaperone WDR79/TCAB1 (*Tycowski et al., 2009*; *Venteicher et al., 2009*). The H/ACA region is required for the correct trafficking, stability, and catalytically active conformation of hTR (*Chen et al., 2018*; *Jády et al., 2004*; *Mitchell et al., 1999a*; *Zhu et al., 2004*) but is considered non-functional as a pseudouridylation guide RNA (*Meier, 2005*).

**Figure 1.** Characterization of the hTR-RNA interactome. (**A**) Schematic representation of hTR sequence, domain organization and bait oligonucleotides (ODNs) used in this study. (**B**) List of top 12 high-confidence hTR interacting RNAs in VA13-hTR cells, ranked based on peak score (JAMM software) across hTR pull-downs. A full list is provided in *Figure 1—source data 1*. Predicted interaction sites in hTR for the top 12 RNAs are shown in panel **A** (blue lines; numbers indicate the rank of the transcript as shown in **B**). Details for these predicted interactions are provided in *Figure 1—source data 2*. (**C**) Overlap between hTR-interacting RNAs identified in VA13-hTR and HeLa cells. A list of interacting partners identified in both cell lines is shown next to the Venn diagram.

DOI: https://doi.org/10.7554/eLife.40037.002

The following source data and figure supplements are available for figure 1:

**Source data 1.** List of hTR interacting RNAs in VA13-hTR and HeLa cells, ranked based on peak scores.

*Figure 1 continued on next page*

*Figure 1 continued*

DOI: https://doi.org/10.7554/eLife.40037.005

**Source data 2.** Details of the predicted RNA-RNA interactions for the transcripts listed in *Figure 1B*.

DOI: https://doi.org/10.7554/eLife.40037.006

**Figure supplement 1.** Schematic pipeline of the experimental protocol employed for the characterization of the hTR-RNA interactome.

DOI: https://doi.org/10.7554/eLife.40037.003

**Figure supplement 2.** Verification of selected hTR-RNA interactions by qRT-PCR.

DOI: https://doi.org/10.7554/eLife.40037.004

Interestingly, while hTERT expression is silenced in most human somatic cells, hTR is broadly expressed in normal tissues (*Feng et al., 1995*). In addition, hTR levels are in excess over telomerase RNP complexes in cancer cells (*Xi and Cech, 2014*), indicating that a pool of TERT-free hTR might assemble into alternate RNP complexes both in normal and transformed cells. Indeed, role(s) independent of telomerase activity – with as yet poorly defined mechanism(s) – have been demonstrated for hTR in cell survival and in the regulation of apoptosis (*Gazzaniga and Blackburn, 2014*; *Kedde et al., 2006*; *Li et al., 2004*). The cell protective function of hTR was mapped to the 3' H/ACA domain and was shown to be negatively regulated by the formation of catalytically active telomerase RNP complexes (*Gazzaniga and Blackburn, 2014*).

Besides serving as flexible scaffolds for protein binding and RNP assembly, most non-coding RNA classes engage in complementarity-driven base-pairing with other RNAs or DNA (*Falaleeva and Stamm, 2013*; *Tay et al., 2014*). In addition, RNA duplex formation has also been suggested to regulate messenger RNA localization/compartmentalization (*Langdon et al., 2018*). In recent years, several methods for either targeted (*Engreitz et al., 2014*; *Kretz et al., 2013*) or transcriptome-wide (*Aw et al., 2016*; *Lu et al., 2016*; *Nguyen et al., 2016*; *Sharma et al., 2016*) mapping of RNA-RNA interactions have been reported, providing the first glimpses into the intricate RNA interaction network in human cells. Although transcriptome-wide methods, relying on psoralen photo-crosslinking and proximity ligation (*Aw et al., 2016*; *Lu et al., 2016*; *Sharma et al., 2016*) have provided important insights into the overall network topology of the RNA interactome, the overlap in the identified interactions using the different methods is rather limited (*Gong et al., 2018*), suggesting that the current approaches might cover only a fraction of the complex cellular RNA interaction network. In addition, psoralen-based methods are also biased by the sequence- and structural features of RNA duplexes, as they preferentially detect interacting regions with staggered uridine bases on opposing strands (*Cimino et al., 1985*; *Lu et al., 2016*).

While the protein composition (*Nguyen et al., 2018*) and chromatin-binding sites (*Chu et al., 2011*) of the telomerase RNP have been characterized in detail, virtually nothing is currently known about RNAs potentially interacting with hTR. In order to better understand the regulation of hTR metabolism and to gain insights into its extra-telomeric role(s), we mapped the RNA interactome of hTR in human cells by a targeted RNA pull-down approach (*Engreitz et al., 2014*), and uncovered a hTR-histone 1C mRNA axis involved in the regulation of human telomere homeostasis.

## Results

### Mapping of the hTR-RNA interactome

We mapped the hTR-RNA interaction network by formaldehyde cross-linking followed by RNA antisense purification and RNA sequencing (RAP-RNA[FA] RNA-seq) (*Engreitz et al., 2014*; the experimental pipeline is shown in *Figure 1—figure supplement 1*). Since telomerase RNP formation is expected to compete with (some of) the alternative functions of hTR (*Gazzaniga and Blackburn, 2014*; *Xi and Cech, 2014*) and can also influence hTR trafficking (*Tomlinson et al., 2008*), we used both hTR−/hTERT− VA13 cells transiently transfected by hTR (VA13-hTR) and hTR+/hTERT+ HeLa cells for hTR antisense purification (*Figure 1—figure supplement 1*). Control RAP-RNA[FA] pull-down of U2 small nuclear RNA (snRNA), as well as mock pull-down from untransfected (hTR-negative) VA13 cells was carried out in parallel. RNA fragments co-purifying with hTR were identified by determining their enrichment in pull-down *vs* input samples. To build a high-confidence set of hTR interacting RNA molecules, only highly (>4 fold) enriched, reproducibly identified peaks were considered

further, resulting in 80 RNA species in VA13-hTR cells. Unfiltered peak calling results produced by the JAMM universal peak finder (*Ibrahim et al., 2015*) are provided in *Supplementary file 1*; the top 12 hTR interacting RNAs are shown in *Figure 1B*, while the full list is provided as *Figure 1— source data 1*.

As expected, the stringent filtering criteria resulted in fewer hTR-interacting RNAs in the TERT[+] HeLa cells (16 RNA species (*Figure 1—source data 1*), out of which 11 were also enriched in pull-downs from VA13-hTR cells; *Figure 1C*), in agreement with a possible competition between active telomerase RNP formation and non-canonical interactions (*Gazzaniga and Blackburn, 2014*).

Although RAP-RNA[FA] can detect both indirect interactions and direct RNA-RNA interactions caged or flanked by proteins (*Engreitz et al., 2014*), prediction of potential duplex formation between hTR and the enriched RNA regions – compared to either the corresponding antisense or shuffled RNA sequences – suggested that the majority of the interactions are mediated by direct RNA-RNA base pairing (*Figure 2A*). Interestingly, the predicted interaction sites fall mostly within regions of hTR that are not thought to be involved in the regulation of telomerase activity or trafficking (*Figure 2B*; indicated in grey in *Figure 1A*), suggesting that these sequences might function as 'hubs' for RNA-RNA interactions.

Confirming the validity of our approach, the stringently filtered dataset included *HSP90AB1*, the only hTR-interacting mRNA identified by the transcriptome-wide LIGR-seq method (*Sharma et al., 2016*). Furthermore, enrichment of selected candidates, such as *TPT1*, *FLNA*, and *IFITM3* was successfully verified by qRT-PCR on RAP samples (*Figure 1—figure supplement 2*).

## *HIST1H1C* RNA specifically interacts with hTR

We identified the *HIST1H1C* transcript, coding for the H1.2 linker histone subtype, as one of the most highly enriched RNAs upon hTR pull-down both in VA13-hTR and HeLa cells. Cell-cycle-regulated histone transcripts are processed in histone locus bodies (HLBs), nuclear structures formed at the sites of histone gene transcription and concentrating factors involved in histone pre-mRNA recognition and maturation (*Nizami et al., 2010*). Although HLBs are highly dynamic in space and time, they generally co-localize with CBs, operationally defined as coilin-positive nuclear foci (*Bongiorno-Borbone et al., 2008*; *Machyna et al., 2014*; *Nizami et al., 2010*). Interestingly, hTR has also been shown to accumulate in CBs throughout the cell cycle (*Jády et al., 2004*; *Zhu et al., 2004*), and to be recruited to telomeres specifically in S phase (*Jády et al., 2006*; *Tomlinson et al., 2006*).

Based on their shared subnuclear localization, cell-cycle-specific regulation, and the specific, high enrichment of *HIST1H1C* in hTR pull-down samples (*Figure 3A and B*), we decided to characterize the *HIST1H1C*-hTR interaction and its potential functional consequences in detail. Prediction of potential base-pairing between *HIST1H1C* and hTR identified a 15-nt long region in the open reading frame (ORF) of *HIST1H1C* (nts 334–348) complementary to the terminal stem-loop sequence of the P6b region of hTR (*Figure 3C*), suggesting a direct RNA-RNA interaction between the two RNAs. We named this 15-nt long region TRIAGE, for telomerase RNA interacting genetic element. The recently published cryo-EM structure of human telomerase RNP (*Nguyen et al., 2018*) indicated that the P6b region is exposed and accessible in the holoenzyme.

In order to verify this RNA-RNA interaction, various mutations disrupting the TRIAGE-P6b complementarity were introduced in hTR (*Figure 3D*). For the ΔP6b variant, the entire terminal stem-loop of the P6b region of hTR was deleted, while for the SW variant we swapped the opposing strands of the terminal stem structure (*Figure 3D*). The RS ('rescue') variant of hTR was designed to disrupt base-pairing with the TRIAGE sequence while introducing complementarity to another region of *HIST1H1C* (nts 91–106; shown in *Figure 3B*). All hTR variants could be expressed in VA13 cells at levels similar to wt hTR (*Figure 3D* uppermost panel), and all could be purified by RAP-RNA[FA] with similar efficiencies (*Figure 3D*). Importantly, enrichment of *HIST1H1C* upon hTR pull-down was abrogated in all P6b mutants (*Figure 3D*), including hTR-RS, indicating that sequence complementarity is necessary but not sufficient for the specific RNA-RNA interaction between the P6b stem-loop and the *HIST1H1C* transcript. In agreement with this, *HIST1H1B* mRNA contains the exact TRIAGE sequence (*Figure 3C*) but was enriched upon hTR pull-down to a much lesser extent than its *HIST1H1C* paralog (*Figure 3A*), suggesting that features besides base complementarity (e.g. secondary structure of the mRNA, specific protein binding *etc.*) determine the interaction.

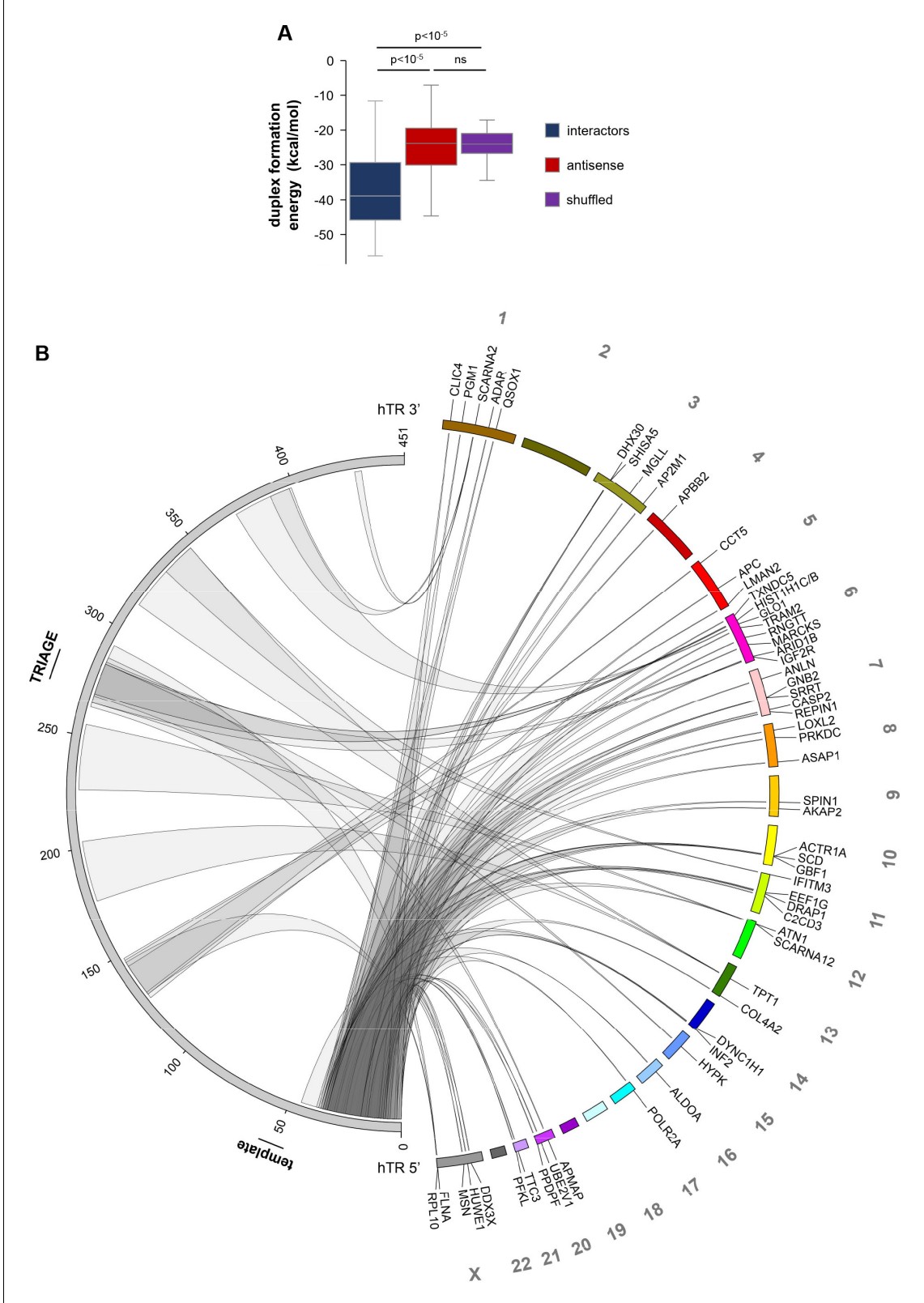

**Figure 2.** Predicted direct hTR-RNA interactions. (**A**) Prediction of duplex formation energies between hTR and RNA sequences enriched in hTR pull-downs in VA13-hTR cells. Antisense and randomly shuffled (5/each RNA) sequences were used as controls representing non-interacting sequences. Statistical analysis was carried out using the Mann-Whitney U test. (**B**) Circos plot (*Krzywinski et al., 2009*) showing the position of predicted direct hTR-RNA interactions. Only interactions with predicted duplex formation energies at least one standard deviation below the median of shuffled

*Figure 2 continued on next page*

*Figure 2 continued*

sequences were included on the plot, corresponding to 58 RNAs (72.5%) out of the 80 RNAs. The left side of the plot corresponds to the hTR sequence (with the position of the template and TRIAGE regions indicated), while the right side represents the genomic position of hTR-RNA interactors.
DOI: https://doi.org/10.7554/eLife.40037.007

## Mutations in hTR disrupting the interaction with *HIST1H1C* RNA lead to increased telomere elongation

The TRIAGE-P6b interaction could lead to changes in the metabolism, activity, or localization of either *HIST1H1C* RNA or hTR. As H1.2 protein levels were not significantly affected by hTR or variant hTR expression (*Figure 4—figure supplement 1*), we determined whether the *HIST1H1C*-hTR interaction might play a role in the regulation of telomere homeostasis. Three independent, polyclonal HT1080 cell lines were established stably expressing each hTR variant. Endogenous hTR levels in HT1080 cells are more limiting in telomere elongation compared to other widely used cancer cell lines, enabling the characterization of mutant hTRs (*Cristofari and Lingner, 2006*). U1 promoter alone (pBS-U1 pr) was used as a negative control, while over-expression of wild-type hTR served as a positive control. We also included the well-characterized m1 hTR mutant (containing the G414C mutation), which fails to accumulate in CBs and causes reduced telomere lengthening in HT1080 cells (*Cristofari et al., 2007*; *Jády et al., 2004*). All hTR variants could be stably over-expressed, resulting in ~3–4 fold hTR expression over endogenous levels (*Figure 4A*). Telomerase activity was measured at day 32 post-transfection using the qTRAP (real-time quantitative telomeric repeat amplification protocol) assay. While deletion of the terminal stem-loop of the P6b region strongly reduced telomerase activity, hTR variants that conserved the overall RNA secondary structure (SW and RS) showed telomerase activities comparable to wild-type hTR, indicating that neither the sequence of the P6b stem-loop nor its interaction with *HIST1H1C* RNA are required for catalytically active telomerase RNP assembly (*Figure 4B*). This is in agreement with a previous report, where substitution of the terminal stem-loop for the GAAA tetraloop sequence showed no influence on telomerase enzymatic activity (*Mitchell and Collins, 2000*). The enzymatic activity associated with hTR variants measured in cell lysates reflects the assembly of telomerase RNPs but – owing to potential trafficking defects – does not necessarily correlate with productive telomere elongation, as previously shown for hTR m1 (*Cristofari et al., 2007*).

In order to examine whether hTR association with *HIST1H1C* RNA affects telomere maintenance, we carried out telomere restriction fragment (TRF) length analysis to follow the mean telomere length (MTL) over time in HT1080 cell lines stably expressing the hTR variants. Expression of hTR variants did not change the cell growth characteristics (*Figure 4C*). As expected, wild-type hTR over-expression resulted in a steady increase in MTL (averaging ~82 nt per population doubling), while U1 promoter alone, the catalytically compromised ΔP6 hTR variant or m1 hTR expression did not result in significant telomere elongation (*Figure 4D and E*, and *Figure 4—figure supplement 2*). Surprisingly, cell lines over-expressing either the SW or RS variant displayed markedly increased telomere lengthening compared to wild-type hTR (*Figure 4D and E*, and *Figure 4—figure supplement 2*), suggesting that the identified *HIST1H1C*-hTR interaction might interfere with the telomeric activity of the telomerase RNP.

## *HIST1H1C* regulates telomere length as a non-coding RNA

If the TRIAGE-P6b interaction inhibits telomere elongation, over-expression of the *HIST1H1C* transcript, independently of its coding potential, could have a negative impact on telomere length maintenance. To examine this possibility, we cloned the *HIST1H1C* coding region, flanked by an N-terminal 3xFLAG tag and the endogenous 5' and 3' regulatory sequences imparting S-phase-specific expression and polyA-independent processing (*Osley, 1991*) into an expression vector (wtHIST1H1C). Several silent codon changes were introduced in the TRIAGE region, disrupting complementarity to the P6b stem-loop sequence of hTR (*Figure 5A*), but maintaining its coding potential (silentHIST1H1C). Three independent, polyclonal HT1080 cell lines were established stably expressing each HIST1H1C construct. We achieved moderate over-expression of 3xFLAG-H1.2 compared to endogenous H1.2 levels, with similar expression levels for 3xFLAG-H1.2 encoded by the wild-type HIST1H1C sequence and the silentHIST1H1C variant (*Figure 5B*). While cancer cells

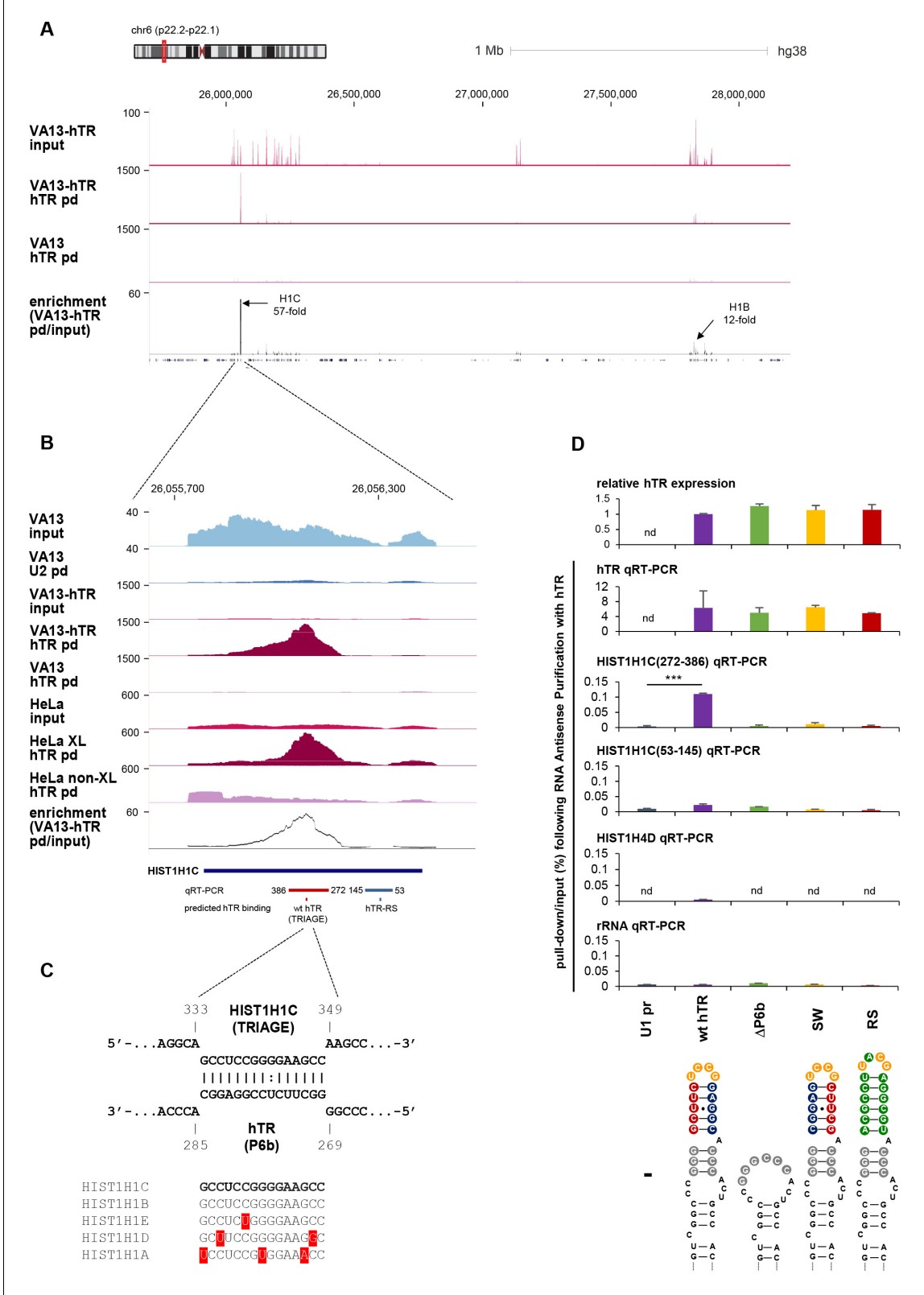

**Figure 3.** *HIST1H1C* RNA specifically interacts with hTR. (**A**) UCSC genome browser view showing coverage of RAP-RNA[FA] RNA-seq over the entire HIST1 gene cluster on chromosome 6 (y axis indicates reads per million). VA13: mock-transfected VA13 cells; VA13-hTR: U1-hTR transfected VA13 cells; pd: pull-down. The position of the *HIST1H1C* and *HIST1H1B* genes and their maximum enrichment upon hTR pull-down is shown. (**B**) Blow-up of the *HIST1H1C* region, showing specific enrichment upon hTR pull-down. Control pull-downs [for U2 snRNA (U2 pd) and hTR pull-down without

*Figure 3 continued on next page*

Figure 3 continued

formaldehyde cross-linking (non-XL)] are also shown. (C) Predicted base-pairing between the TRIAGE sequence and the P6b stem-loop of hTR. The conservation of the TRIAGE sequence in the five replication-dependent somatic linker histone subtypes (*HIST1H1A-E*) is shown below. (D) Mutations were introduced into hTR (as shown at the bottom), disrupting complementarity with TRIAGE (SW: swap mutant; RS: rescue mutant). Relative expression levels of hTR variants (uppermost panel) and pull-down efficiencies of various transcripts (all other panels) were measured by qRT-PCR upon transient transfections of VA13 cells with the indicated hTR variants and RAP-RNA[FA] using hTR-specific antisense oligonucleotides. The results demonstrate the specific pull-down of *HIST1H1C* by wild-type hTR. *HIST1H4D* and ribosomal RNA were used as negative controls. nd: not detectable. The positions of the regions amplified for *HIST1H1C* are illustrated in panel B. Error bars represent s.d. Representative results from two biological replicates, measured in triplicates, are shown. Paired two-tailed *t*-tests, ***p<0.001.
DOI: https://doi.org/10.7554/eLife.40037.008

maintain a remarkable telomere length equilibrium over a long period ex vivo (*Figure 4D and E*), over-expression of wtHIST1H1C resulted in telomere attrition and accumulation of short telomeres in all three cell lines after 34 population doublings (*Figure 5C*). Importantly, silentHIST1H1C expression had no effect on the MTL (*Figure 5C*), demonstrating that the presence of the TRIAGE sequence is responsible for the observed phenotype, possibly regulating telomere length via the *HIST1H1C*-hTR RNA-RNA interaction, independently of H1.2 protein expression. *HIST1H1C* over-expression had no influence on telomerase enzymatic activity measured in cell lysates (*Figure 5E*), suggesting that the TRIAGE sequence is affecting a step following telomerase RNP assembly.

To test whether *HIST1H1C* exerts its effect via a direct RNA-RNA interaction, we carried out a rescue experiment by introducing mutations in the P6b stem-loop of hTR, restoring complementarity to the mutated TRIAGE sequence in silentHIST1H1C (P6b_sil hTR variant; *Figure 5—figure supplement 1A*). P6b_sil hTR could be expressed both in HT1080 and VA13 cells, albeit at levels ~ 4–5 fold lower than wt hTR (*Figure 5—figure supplement 1B and C*). We attribute this difference to a potential reduction in hTR stability due to the disruption of RNA secondary structure. In addition to its defective cellular accumulation, the P6b_sil variant also lost telomerase enzymatic activity (*Figure 5—figure supplement 1B*), similarly to the ΔP6b variant with misfolded P6b stem-loop structure.

Various combinations of hTR and *HIST1H1C* constructs – with or without the potential to form a TRIAGE-P6b RNA duplex – were co-transfected into VA13 cells, followed by RAP-RNA[FA] and qRT-PCR to assess *HIST1H1C*-hTR interaction. As shown in *Figure 5—figure supplement 1C and D*, the P6b_sil variant could efficiently pull down *HIST1H1C* from silentHIST1H1C-transfected cells, but not from wtHIST1H1C-transfected cells. By digesting the PCR products, using the NciI restriction enzyme that specifically cleaves the wtHIST1H1C amplicon, we could show that wt hTR specifically interacted with wild-type *HIST1H1C*, while P6b_sil hTR only interacted with the silentHIST1H1C transcript, demonstrating a direct, complementarity-driven interaction between the two RNAs (*Figure 5—figure supplement 1D*).

Continued telomere attrition in human cells is expected to eventually result in sustained DNA damage signaling and cellular senescence (*Maciejowski and de Lange, 2017*). Strikingly, two of the three polyclonal wtHIST1H1C-expressing cell lines, but none of the silentHIST1H1C-expressing cell lines, lost transgene expression upon long-term culture (assessed at 120 days post-transfection; *Figure 5D*). This strong selection pressure prevented us from reliably assessing the influence of TRIAGE-mediated telomere shortening on cell physiology in this model system, and suggested that cells without (or with very low levels of) H1.2 over-expression might possess a substantial survival advantage. In agreement with this, cell line 1 – maintaining wild-type H1.2 over-expression (*Figure 5D*, lane 4) – exhibited increased p53 protein levels, indicative of DNA damage response activation in this cell population.

To investigate the long-term consequences of *HIST1H1C* RNA over-expression on telomere homeostasis and cell physiology, we followed two complementary strategies. First, we generated 24 clonal HT1080 cell lines stably expressing the wtHIST1H1C construct. Although most clones expressed detectable levels of 3xFLAG-H1.2 protein (*Figure 6—figure supplement 1A*), only 12 out of 24 clones showed >20% upregulation in their *HIST1H1C* RNA levels (*Figure 6A*) relative to the baseline expression measured in mock-transfected cells.

To obtain more sensitive telomere length measurements, we took advantage of the recently developed Telomere Shortest Length Assay (TeSLA) method that facilitates the identification of critically shortened telomeres (*Lai et al., 2017*). *HIST1H1C* over-expression in the clonal cell lines was

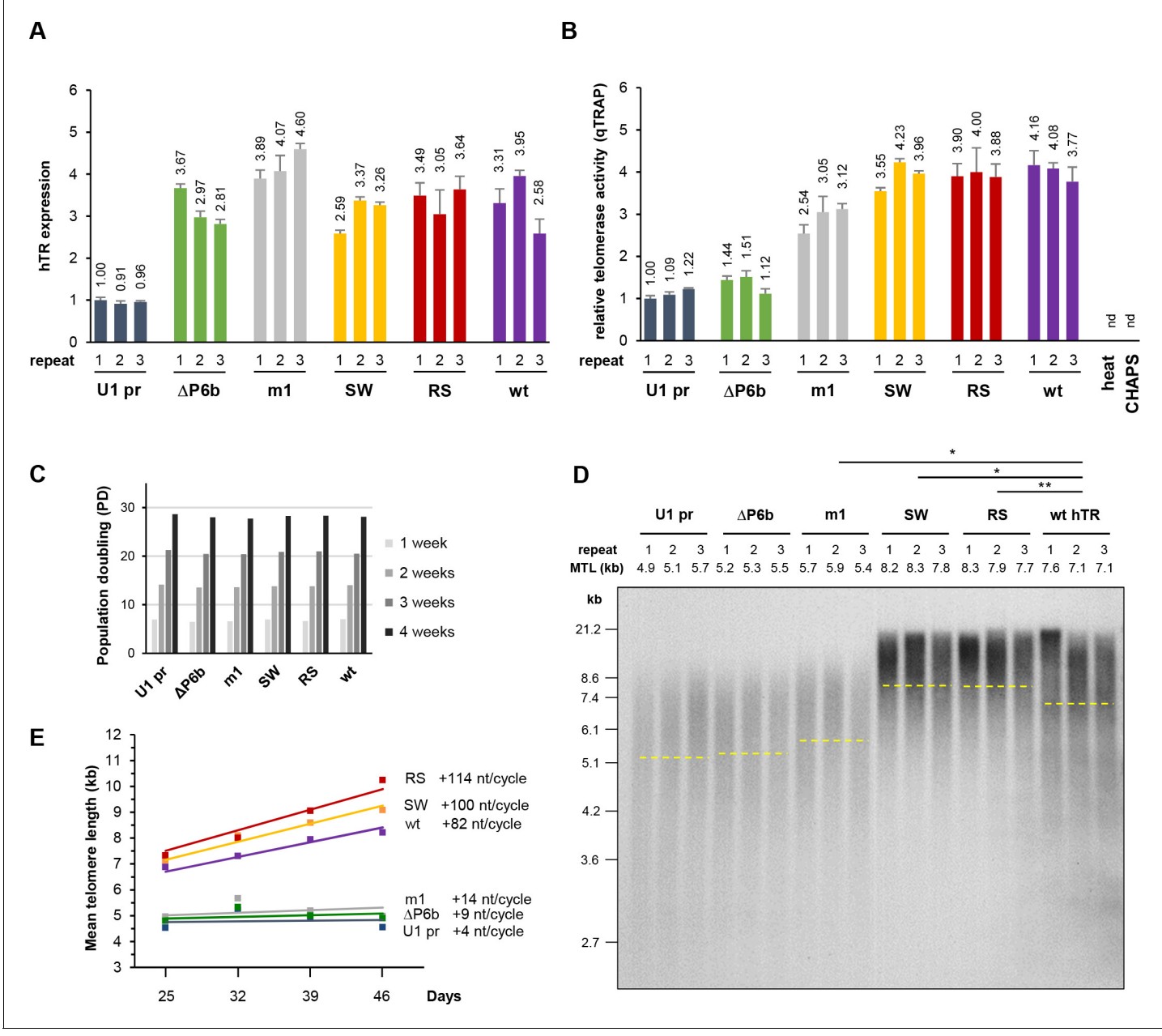

**Figure 4.** Mutations in hTR disrupting the interaction with *HIST1H1C* RNA lead to increased telomere elongation. (A) Relative hTR expression levels and (B) relative telomerase activities in stably transfected polyclonal HT1080 cell lines expressing the indicated hTR variants, measured at day 32 post-transfection. Error bars represent s.d. Representative results of two-to-three biological replicates, measured in triplicates, are shown. (C) Cells were counted at each passage to determine average population doubling (PD) times for the HT1080 cell lines (n = 3 for each condition) stably expressing the indicated hTR variant. (D) Southern blot for TRF analysis at day 32 post-transfection. MTLs for the individual cell lines are indicated above the panel, while the average MTL for each hTR variant is shown by dashed lines. Statistical significance was calculated by paired two-tailed *t*-tests, *p<0.05, **p<0.01. Southern blots for the other time points are provided in *Figure 4—figure supplement 2*. (E) MTL changes over time for HT1080 cells stably expressing the indicated hTR variants. Average MTL values from three independent cell lines are shown. The intercept at day 0 for the trendlines was set at the MTL of the untransfected cell population.

DOI: https://doi.org/10.7554/eLife.40037.009

The following figure supplements are available for figure 4:

**Figure supplement 1.** H1.2 protein levels are not affected by hTR expression.
DOI: https://doi.org/10.7554/eLife.40037.010

**Figure supplement 2.** Southern blots from different time-points post-transfection for telomere length analysis of stably transfected HT1080 cell lines over-expressing the hTR variants.

*Figure 4 continued on next page*

*Figure 4 continued*

DOI: https://doi.org/10.7554/eLife.40037.011

generally associated with shorter MTL and a higher frequency of short telomeres (*Figure 6B and C*, red and orange lines and circles and *Figure 6—figure supplement 2*). In agreement with the accumulation of shorter telomeres, we observed an increase in cells undergoing cellular senescence, as measured by β-galactosidase staining at two time points (8 weeks and 10 weeks post-transfection) (*Figure 6D and E*, red and orange circles). Interestingly, hTR expression did not correlate with cellular senescence (*Figure 6D* and *Figure 6—figure supplement 1*), suggesting that clonal variations in hTR levels are not limiting for the maintenance of the shortest telomeres in this setting.

Secondly, we introduced a frameshift at the N-terminus of the *HIST1H1C* ORF (*Figure 7A*), to characterize the consequences of elevated wtHIST1H1C or silentHIST1H1C RNA expression levels without the potential confounding effects of H1.2 protein over-expression (*Figure 7B*). Three independent, polyclonal HT1080 cell lines were established stably expressing each frameshifted (FS) *HIST1H1C* variant. All cell lines were found to over-express *HIST1H1C* mRNA in a sustained manner (*Figure 7C*). Although there was no statistically significant difference in MTL between the FS_HIST1H1C and FS_silentHIST1H1C-transfected cells at 1 month post-transfection (p=0.62), by 3 months, all cell lines over-expressing FS_HIST1H1C (with wild-type TRIAGE sequence) had significantly shorter telomeres (on average by ~700 nt, p=0.011) than the FS_silentHIST1H1C-transfected controls (*Figure 7D* and *Figure 7—figure supplement 1*).

Taken together, our experiments with polyclonal and clonal HT1080 cell lines demonstrate a role for *HIST1H1C* RNA in telomere length homeostasis and cell survival. Since the TRIAGE-P6b interaction does not affect telomerase enzymatic activity (*Figures 4B* and *5E*), *HIST1H1C* RNA might act at a step following telomerase RNP assembly, possibly as a telomerase sponge preventing the recruitment of the enzymatically active telomerase complex to telomeres.

## The hTR-RNA interactome and extra-telomeric functions of hTR

In addition to *HIST1H1C*, we identified a multitude of hTR-interacting RNA partners that might provide interesting new insights into the extra-telomeric function(s) of hTR. Strikingly, functional annotation analysis (*Huang et al., 2009a*; *Huang et al., 2009b*) showed that 35 out of 77 (45.5%) hTR-interacting mRNAs identified in VA13-hTR cells code for proteins involved in cytoskeleton organization and/or the regulation of apoptosis (*Figure 8*), thus providing a direct link – mediated by an intricate RNA-RNA interaction network – between hTR expression and cell survival (*Gazzaniga and Blackburn, 2014*). Prominent hTR-associated mRNAs coding for apoptotic factors include the translationally controlled tumor protein (TCTP/TPT1) and filamin A (FLNA) RNAs (*Figure 1—figure supplement 2*), ranked sixth and first in our screen, respectively (*Figure 1B*). The two proteins interact with each other and are important pleiotropic regulators of DNA repair, apoptosis, tumorigenesis, and cytoskeleton organization (*Amson et al., 2013*; *Zhang et al., 2012*). Notably, TPT1 interacts with and regulates various apoptotic factors, including Bcl-X$_L$ (*Thébault et al., 2016*; *Yang et al., 2005*) and Mcl1 (*Liu et al., 2005*; *Zhang et al., 2002*), two regulators of the Bim apoptotic pathway, which was reported to be inhibited by hTR expression (*Gazzaniga and Blackburn, 2014*).

In addition, hTR selectively interacts with a small subset of scaRNAs – most prominently with scaRNA2 and scaRNA17 – in both VA13-hTR and HeLa cells (*Figure 9A–C*). As expected for a specific interaction, the association with scaRNAs was significantly reduced with the m1 hTR variant (*Cristofari et al., 2007*; *Jády et al., 2004*) that does not accumulate in CBs (*Figure 9C*). By predicting base-pairing interactions between hTR and the enriched scaRNAs, we found that scaRNA2 can potentially occupy both pseudouridylation pockets of hTR, leaving an unpaired uridine exposed (U27 and U34 for the 5' and 3' pockets, respectively) (*Figure 9D*). This suggests the exciting possibility that hTR might behave as an active pseudouridylation guide RNA.

## Discussion

Using a targeted RNA pull-down approach (*Engreitz et al., 2014*), we have identified ~80 RNA species interacting directly or indirectly with hTR. In agreement with the previously published anti-apoptotic role of hTR (*Gazzaniga and Blackburn, 2014*), our dataset is highly enriched in mRNAs

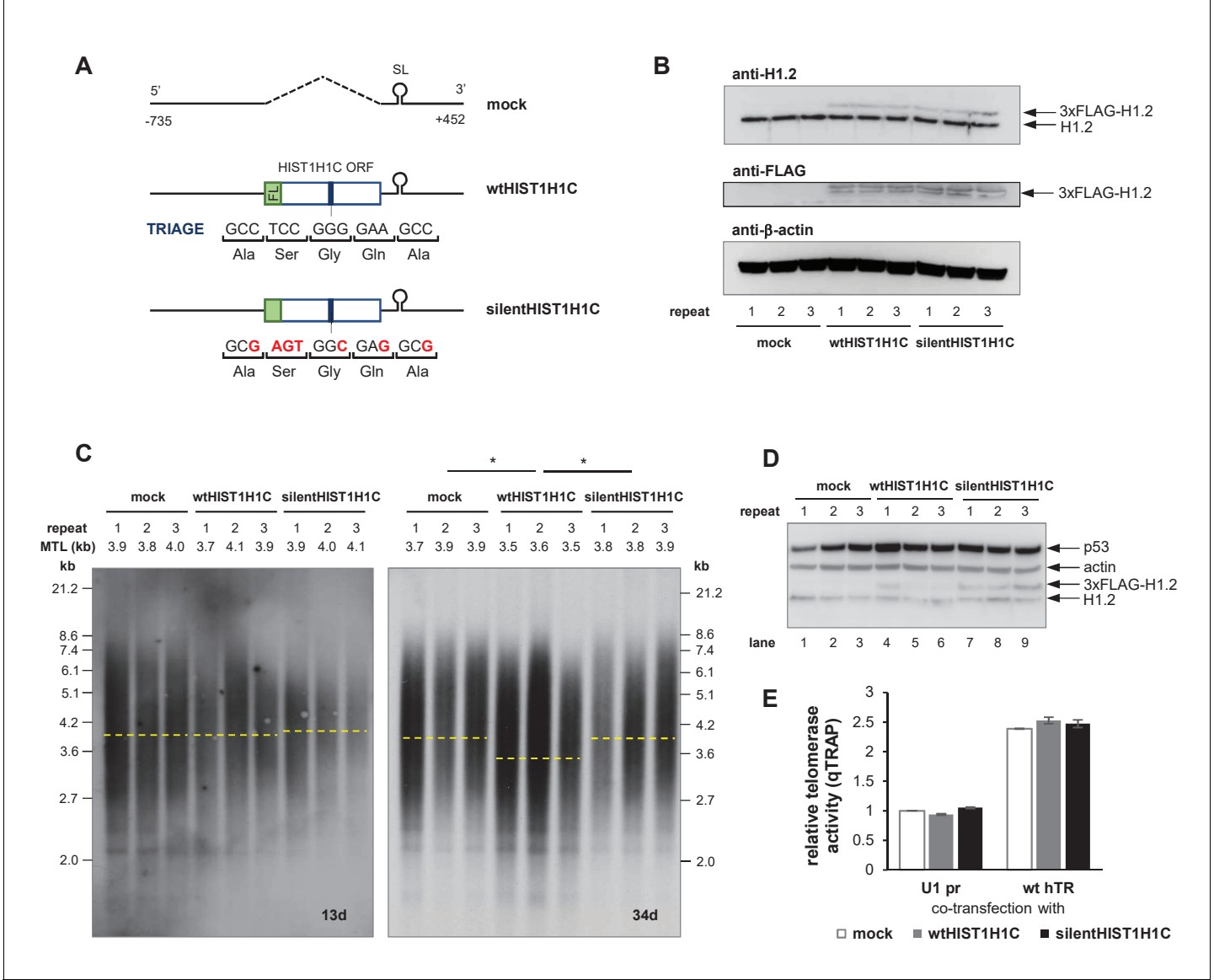

**Figure 5.** The presence of the TRIAGE sequence in *HIST1H1C* regulates telomere length. (**A**) Schematic representation of HIST1H1C constructs. The TRIAGE sequence and silent mutations introduced in it are shown. FL indicates the N-terminal 3xFLAG tag. (**B**) Western blot showing 3xFLAG-H1.2 expression in stable polyclonal HT1080 cell lines. The same membrane was sequentially probed with anti-H1.2, anti-actin, and anti-FLAG antibodies. A doublet is detected with the anti-FLAG antibody, probably due to a phosphorylated H1.2 form that is not recognized by the anti-H1.2 antibody. (**C**) Southern blots for TRF analysis at days 13 (left panel) and 34 (right panel) post-transfection. MTLs for the individual cell lines are indicated above the panel, while the average MTL with the various HIST1H1C constructs is shown by dashed lines. Statistical significance was calculated by paired two-tailed *t*-tests, *p<0.05. (**D**) Western blot showing the loss of 3xFLAG-H1.2 expression in wtHIST1H1C-expressing polyclonal cell lines 2 and 3 (lanes 5 and 6) upon long-term culture (120 days post-transfection), and activation of the DNA damage marker p53 in cell line 1 (lane 4). (**E**) Relative telomerase activities upon transient transfection of HT1080 cells with the indicated combinations of hTR- (U1 pr and wt hTR) and histone 1C (mock, wtHIST1H1C, and silentHIST1H1C)-expressing constructs, measured 3 days post-transfection. Error bars represent s.d. Representative results of two biological replicates, measured in triplicates, are shown.

DOI: https://doi.org/10.7554/eLife.40037.012

The following figure supplement is available for figure 5:

**Figure supplement 1.** Rescue of the *HIST1H1C*-hTR interaction by compensatory mutations in the two RNAs.

DOI: https://doi.org/10.7554/eLife.40037.013

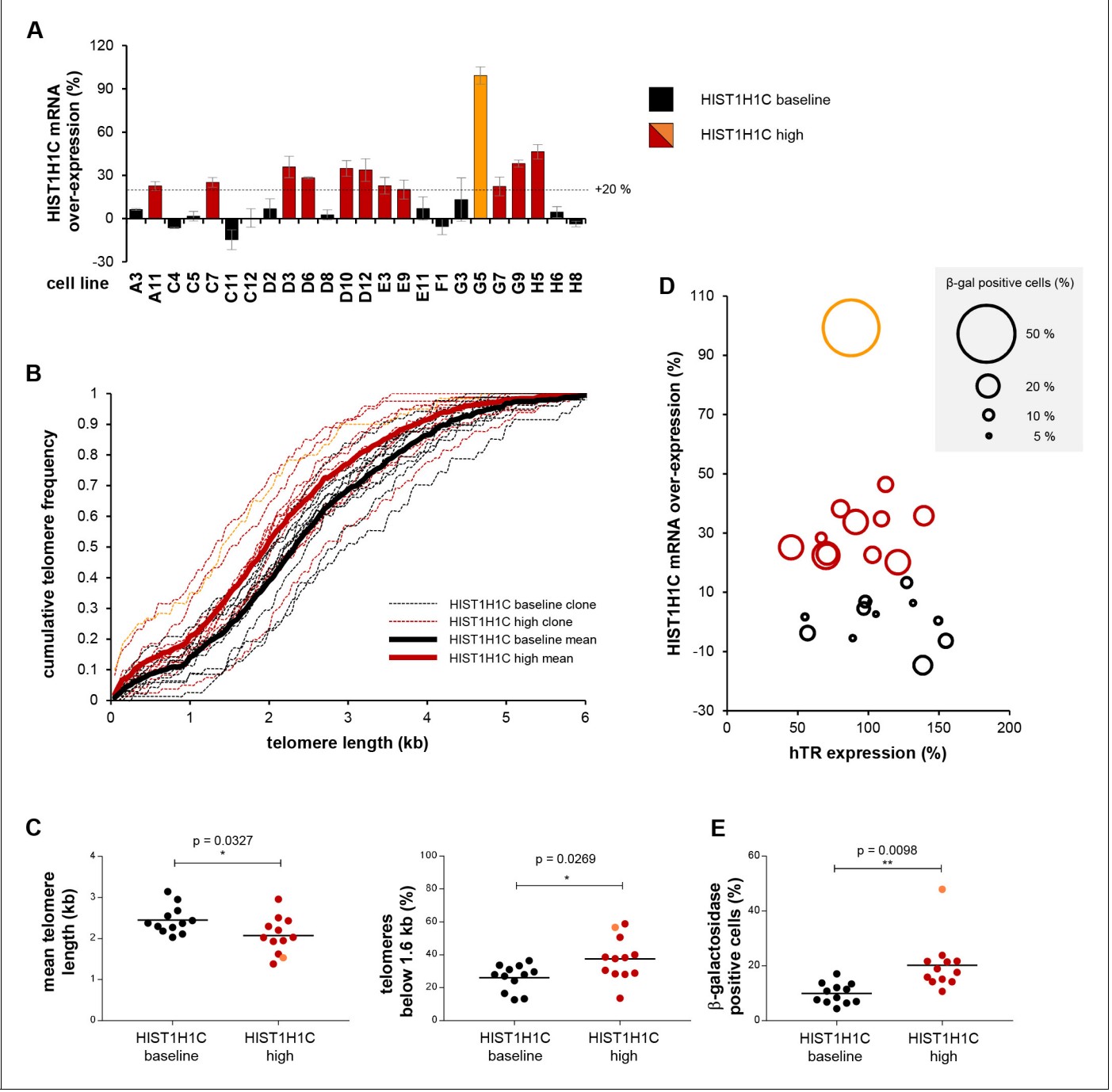

**Figure 6.** Long-term *HIST1H1C* mRNA over-expression results in shorter telomere length and increased cellular senescence. (**A**) *HIST1H1C* mRNA over-expression levels relative to mock-transfected cells in clonal HT1080 cell lines stably transfected with wtHIST1H1C, measured by qRT-PCR at 10 weeks post-transfection. Values were normalized to *HIST1H3B* RNA levels. Cell lines were divided into two groups, each comprising 12 members: clones with <20% over-expression relative to mock (in black; 'HIST1H1C baseline' group) and clones with >20% over-expression (in red and orange; 'HIST1H1C high' group). Clone G5, showing the highest *HIST1H1C* RNA expression levels, is emphasized by orange colour. The same colour code is applied throughout panels B-E and in *Figure 6—figure supplement 1*. Error bars represent s.d., n = 2, with triplicates each. (**B**) Telomere length of clonal HT1080 cell lines was assessed by the TeSLA method (*Lai et al., 2017*) at 10 weeks post-transfection. Cumulative telomere frequency (calculated as the ratio of telomeres below a certain length) is shown for the individual clonal cell lines (thin dashed lines) and for the mean of the 'HIST1H1C baseline' and 'HIST1H1C high' groups (thick lines). TeSLA blots and the associated telomere length values for the individual clones are provided in *Figure 6—figure supplement 2*. (**C**) Elevated *HIST1H1C* mRNA expression levels correlate with lower mean telomere length values (left panel) and with an increase in the ratio of short telomeres (arbitrarily defined as telomeres below 1.6 kb; right panel). Each data point corresponds to a clonal HT1080 cell line stably expressing wtHIST1H1C. Statistical significance was calculated by paired two-tailed *t*-tests, *p<0.05. TeSLA blots and telomere length values

*Figure 6 continued on next page*

*Figure 6 continued*

for the individual clones are provided in *Figure 6—figure supplement 2*. Note that telomere length measurements with the TeSLA method give shorter MTL values than the TRF method utilized in *Figures 4* and *5*, and the absolute values obtained with these distinct methods are not directly comparable (*Lai et al., 2017*). (D) Elevated *HIST1H1C* mRNA expression levels correlate with increased cellular senescence, as measured by β-galactosidase staining. Cellular senescence in individual HT1080 clonal cell lines stably expressing wtHIST1H1C, measured at 8 and 10 weeks post-transfection, is illustrated by the diameter of the bubbles, plotted as a function of hTR expression levels (expressed relative to the average value in the 24 clonal cell lines) and *HIST1H1C* mRNA over-expression levels (relative to mock-transfected cells). (E) Statistical significance between the 'HIST1H1C baseline' and 'HIST1H1C high' groups was calculated by paired two-tailed *t*-tests, **p<0.01.

DOI: https://doi.org/10.7554/eLife.40037.014

The following source data and figure supplements are available for figure 6:

**Source data 1.** Numerical values measured for the clonal HT1080 cell lines.
DOI: https://doi.org/10.7554/eLife.40037.017
**Figure supplement 1.** Western blots, cellular senescence assays, and qRT-PCR measurements on wtHIST1H1C-expressing clonal HT1080 cell lines.
DOI: https://doi.org/10.7554/eLife.40037.015
**Figure supplement 2.** Southern blots and telomere length measurements of clonal, wtHIST1H1C-transfected HT1080 cell lines.
DOI: https://doi.org/10.7554/eLife.40037.016

coding for proteins involved in the regulation of apoptosis, suggesting that (some of) the extra-telomeric functions of hTR might be mediated through RNA-RNA interactions.

Interestingly, only one of the mRNAs enriched upon hTR pull-down in our study (*HSP90AB1*) has been previously reported to interact with hTR in transcriptome-wide screens of cellular RNA-RNA interactions (*Aw et al., 2016*; *Lu et al., 2016*; *Sharma et al., 2016*). This probably reflects both the insufficient saturation of these methods (*Gong et al., 2018*) and the biases resulting from psoralen photo-crosslinking (*Cimino et al., 1985*; *Lu et al., 2016*). Indeed, many putative hTR-RNA duplexes, including the TRIAGE-P6b interaction (*Figure 3C*) and the predicted scaRNA2-hTR interaction (*Figure 9D*), do not contain uridines in a position that would conform to the substrate requirements of psoralen crosslinking. As RNA molecules have an almost unlimited potential to engage in base-pairing interactions and some species accumulate only at single-digit numbers per cell, our results – in agreement with earlier targeted approaches (*Engreitz et al., 2014*; *Kretz et al., 2013*) – suggest that, despite major technological advances in the last decade, a large fraction of the cellular RNA interactome remains unexplored.

Interestingly, in recent years, a growing number of snoRNAs and other non-coding RNAs have been shown to be further processed to various, smaller fragments and thus regulate various cellular functions in addition to their canonical roles (e.g. *Falaleeva and Stamm, 2013*; *Kumar et al., 2016*; *Rogler et al., 2014*). We speculate that some of the interactions identified in our study might, at least partially, reflect specific functions executed by putative, hTR-derived smaller RNA species, a possibility that remains to be addressed experimentally.

Our study identified and characterized *HIST1H1C* RNA as a negative regulator of telomere elongation through its interaction with hTR. Using a combination of hTR variants and *HIST1H1C* mutants containing either silent mutations (disrupting complementarity to hTR without interfering with H1.2 protein expression), or a frameshift at the start of the *HIST1H1C* ORF (abolishing protein production without limiting RNA expression), we demonstrated that the telomere elongation phenotype is mediated by *HIST1H1C* acting as a non-coding RNA, and not through H1.2 protein expression. As neither *HIST1H1C* over-expression, nor mutations in the P6b stem-loop of hTR disrupting *HIST1H1C* association had an effect on telomerase enzymatic activity, we propose that *HIST1H1C* regulates telomere length homeostasis acting at a step downstream of RNP assembly, possibly by sequestering the active telomerase complex and preventing its proper intranuclear targeting or recruitment to telomeres. As both hTR and replication-dependent histone (pre)mRNAs accumulate in Cajal bodies (*Bongiorno-Borbone et al., 2008*; *Machyna et al., 2014*; *Jády et al., 2004*; *Zhu et al., 2004*), the interaction between the two RNAs might possibly take place in these structures. Co-localization experiments between *HIST1H1C* mRNA and hTR, in different stages of the cell cycle, will have to be carried out to gain further insights into the exact mechanisms governing this regulation, which appears to substantially contribute to the exquisitely fine-tuned cellular homeostasis of telomere length maintenance.

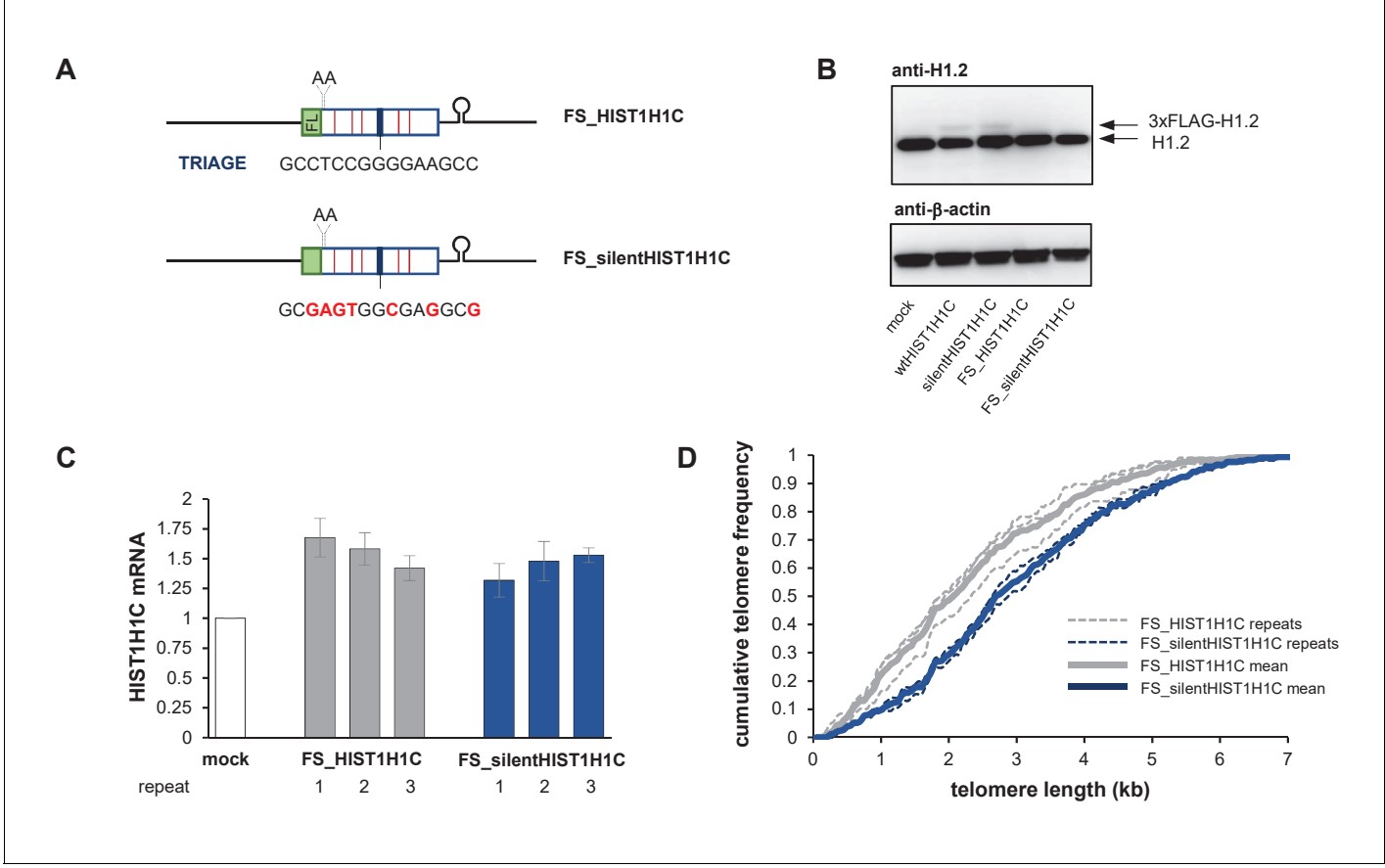

**Figure 7.** *HIST1H1C* regulates telomere length acting as a non-coding RNA. (**A**) Schematic representation of HIST1H1C expression constructs containing a two-nucleotide frameshift achieved by insertion of two adenines following the 3xFLAG tag (FL). The frameshift results in several stop codons in the *HIST1H1C* ORF (red lines), and completely abolishes 3xFLAG-H1.2 expression, assessed by western blotting (**B**). (**C**) *HIST1H1C* mRNA expression levels in polyclonal HT1080 cell lines stably transfected with the indicated constructs were measured 3 months post-transfection by qRT-PCR. Error bars represent s.d., n = 2, with triplicates each. (**D**) Telomere length of the polyclonal HT1080 cell lines was measured by the TeSLA method at 3 months post-transfection. Cumulative telomere frequency (calculated as the ratio of telomeres below a certain length) is shown for the individual cell lines (thin dashed lines) and as average for the three biological replicates (thick lines). TeSLA blots and the associated telomere length values for the individual cell lines are provided in *Figure 7—figure supplement 1*.

DOI: https://doi.org/10.7554/eLife.40037.018

The following figure supplement is available for figure 7:

**Figure supplement 1.** Southern blots and telomere length measurements of polyclonal HT1080 cell lines stably expressing the frameshifted HIST1H1C constructs.

DOI: https://doi.org/10.7554/eLife.40037.019

The human genome encodes 11 histone 1 (H1) subtypes, five of which (*HIST1H1A* to *HIST1H1E*) are located – along with a number of core histone genes – in the large HIST1 gene cluster on chromosome 6, and are expressed in a cell cycle-dependent fashion in somatic cells (*Harshman et al., 2013*; *Hergeth and Schneider, 2015*). H1 subtypes present cell cycle, cell type, and tissue-specific differences in their expression patterns, and their relative ratios undergo profound changes during differentiation, malignant transformation, cancer progression, or under distinct physiological conditions (*Happel et al., 2009*; *Hergeth and Schneider, 2015*; *Izzo et al., 2017*; *Pan and Fan, 2016*; *Scaffidi, 2016*; *Terme et al., 2011*). Interestingly, the TRIAGE sequence is conserved between *HIST1H1C* and *HIST1H1B*, and shows only one nucleotide difference in *HIST1H1E*, maintaining complementarity to hTR through G-U base pairing (*Figure 3C*). Although we observed some enrichment in hTR pull-downs for these two subtypes (and none for *HIST1H1D* with two nucleotides difference in the TRIAGE sequence), we consistently detected a preferential enrichment of *HIST1H1C* mRNA over the other variants. The reason for this specificity is currently unclear, but could potentially be

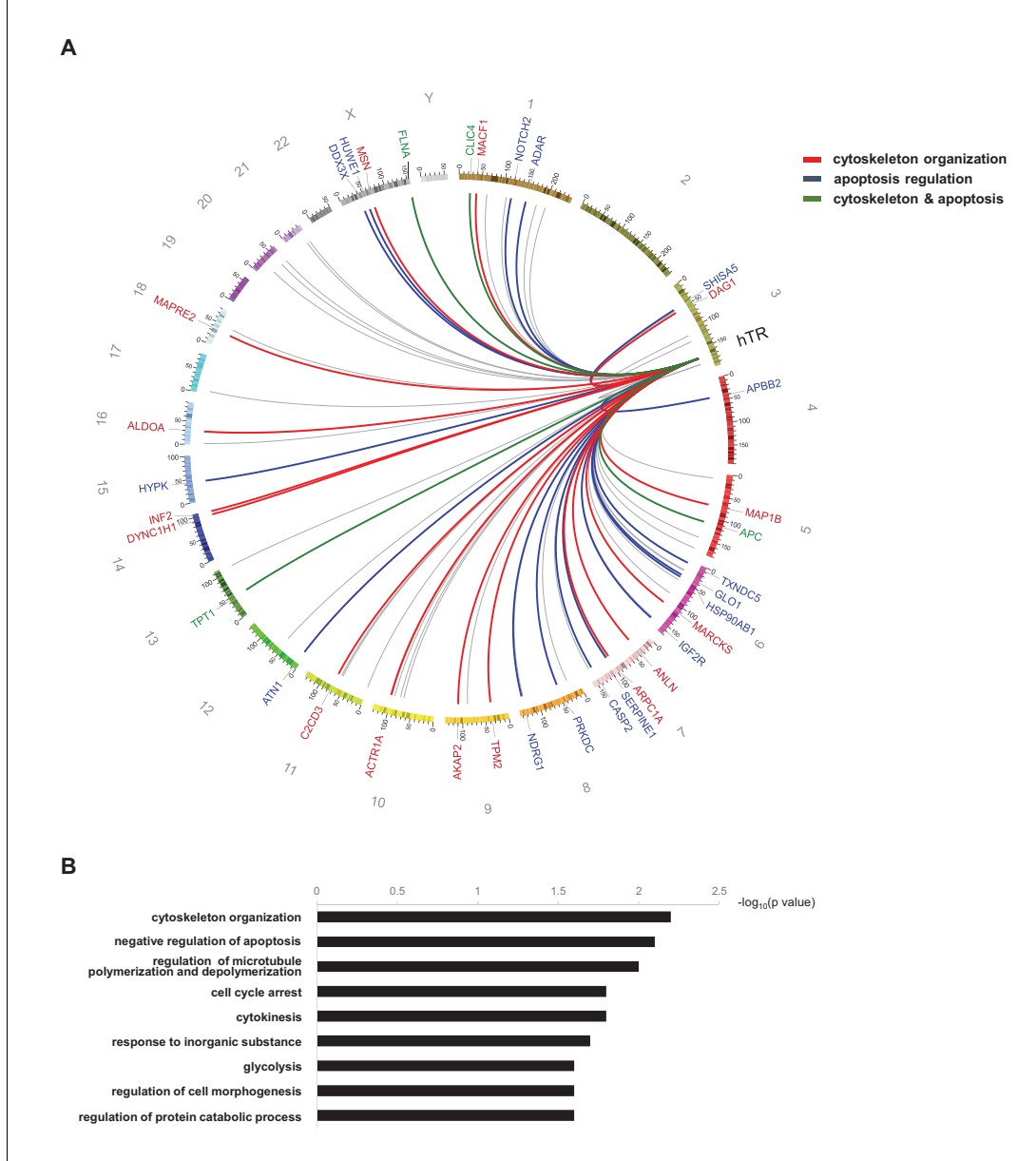

**Figure 8.** Functional annotation analysis of the hTR interactome. (**A**) Circos plot (*Krzywinski et al., 2009*) depicting hTR-interacting mRNAs coding for proteins functionally associated with cytoskeleton organization (red links and labels), apoptosis regulation (blue links and labels), or both (green links and labels). Links shown in grey correspond to RNAs not functionally associated with either cytoskeleton organization or apoptosis. The most significant gene ontology terms (GOTERM_BP_FAT, using DAVID 6.7) associated with the hTR RNA interactome are shown in (**B**).

DOI: https://doi.org/10.7554/eLife.40037.020

explained by different intramolecular RNA secondary structures involving the TRIAGE sequence in *HIST1H1B* and *HIST1H1E*, or by the involvement of an RNA-binding protein – preferentially associating with the *HIST1H1C* mRNA – facilitating the TRIAGE-P6b RNA duplex formation. Nevertheless, it remains possible that under different conditions or in different cell/tissue types *HIST1H1B* and/or *HIST1H1E* might more efficiently interact with hTR and regulate its function(s).

Beyond their role in regulating chromosome condensation as structural constituents of chromatin, linker histones have been implicated in a wide variety of cellular processes (reviewed in *Hergeth and Schneider, 2015*), including the specific regulation of gene expression, apoptosis and the DNA damage response. Seemingly contradictory findings for H1 functions in genome stability have been

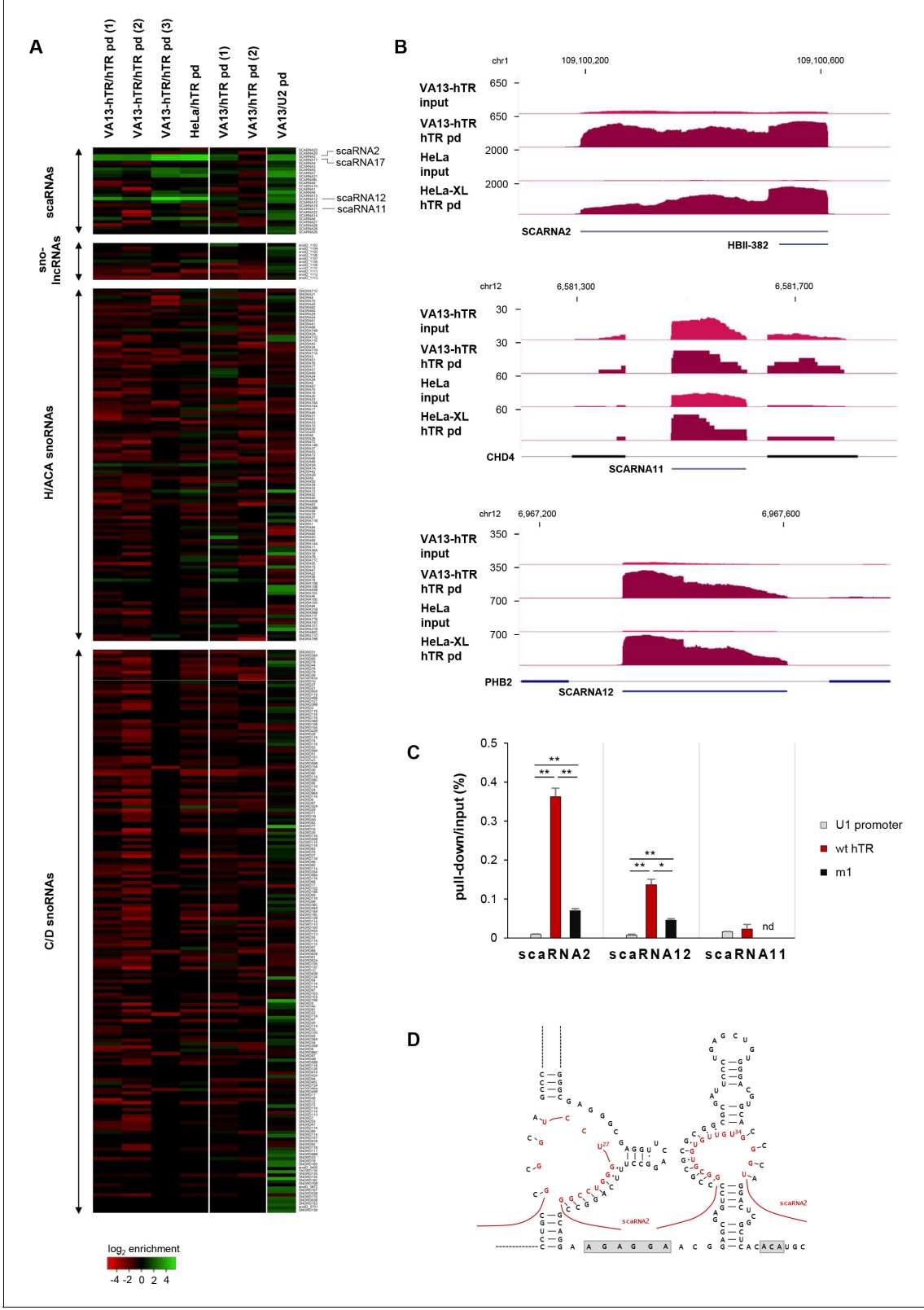

**Figure 9.** hTR selectively interacts with a subset of small Cajal-body-specific non-coding RNAs. (**A**) Heat map showing the enrichment of small nucleolar RNAs (snoRNAs) and small Cajal body-specific RNAs (scaRNAs) upon hTR or U2 small nuclear RNA pull-down (pd). SnoRNA co-ordinates were obtained from *Jorjani et al. (2016)*. Only snoRNAs with at least 10 RNA-sequencing reads supporting their expression were included in the figure. The position of highly enriched transcripts upon hTR pull-down (scaRNA2, scaRNA17, and scarRNA12), as well as the non-enriched scaRNA11 are indicated on the

*Figure 9 continued on next page*

*Figure 9 continued*

right. Figure generated using Heatmapper (*Babicki et al., 2016*). Note that U2 is expected to interact with scaRNAs, as they mediate the modification of snRNAs. (B) UCSC genome browser view showing coverage of RAP-RNA[FA] RNA-seq for selected scaRNAs. The y axis indicates reads per million. (C) qRT-PCR of scaRNA enrichments upon hTR pull-down from VA13 cells transiently transfected with the hTR expression constructs indicated. Error bars represent s.d. Representative results from two independent transfections, measured in triplicates, are shown. nd: not detectable. Paired two-tailed *t*-tests, *p<0.05, **p<0.01. (D) Hypothetical base-pairing between scaRNA2 and the pseudouridylation pockets of hTR. Base-pairing in the 5' pocket will require the formation of an alternative secondary structure for the 5' H/ACA hairpin of hTR (*Egan and Collins, 2010*; *Theimer et al., 2007*), different from the canonical structure (*Chen et al., 2000*) shown in *Figure 1A*.

DOI: https://doi.org/10.7554/eLife.40037.021

reported, both for H1 histones in general and for H1.2 in particular. For example, H1.2 has been shown to repress p53-mediated transcriptional activation of target genes through – in part – a direct interaction with p53 (*Kim et al., 2008*; *Nishiyama et al., 2009*), conferring an anti-apoptotic function upon DNA damage (*Nishiyama et al., 2009*). In contrast, the release of H1.2 from chromatin in response to DNA double-strand breaks has been shown to lead to its translocation to the cytoplasm and to result in the activation of the intrinsic apoptotic pathway and increased cell death in a Bak-dependent manner (*Garg et al., 2014*; *Konishi et al., 2003*). Chromatin compaction states, regulated by linker histone abundance, might also influence various fundamental cellular processes, including the DNA damage response (*Murga et al., 2007*; *Sulli et al., 2012*) and transcription from heterochromatic repeat regions (*Izquierdo-Bouldstridge et al., 2017*), potentially modulating telomere length homeostasis. Murga and co-workers reported two lines of triple knockout mouse embryonic stem cells [TKO mES cells, lacking H1.2, H1.3 and H1.4 expression (*Fan et al., 2003*)] that showed increased telomere length compared to lines expressing wild-type levels of linker histones (*Murga et al., 2007*). These differences might be attributed to an ALT (alternative lengthening of telomeres)-like phenotype due to increased homologous recombination occurring in the relaxed chromatin environment caused by the drastic reduction in H1 histone expression levels (*Murga et al., 2007*). Interestingly, although the sequence of the P6b stem-loop (as well as that of the TRIAGE sequence) is highly conserved in most placental mammals, mouse and rat P6b is highly divergent, suggesting that the P6b-TRIAGE interaction would not occur in these species. Whether hTR can bind to other region(s) of the rodent *Hist1h1c* mRNA (or potentially to other mouse/rat histone mRNAs) remains to be determined. Alternatively, the potential lack of this interaction might reflect divergent evolutionary strategies employed by these short-lived, small species with excessively long telomeres and somatic telomerase activity (*Gomes et al., 2011*). In agreement with distinct telomerase trafficking regulation in human and mouse cells, mouse telomerase RNA does not accumulate in CBs, despite the presence of a canonical CAB box in its scaRNA region (*Tomlinson et al., 2010*).

Collectively, our results support a model where the *HIST1H1C* and hTR RNAs directly interact through the TRIAGE-P6b complementarity, resulting in an attenuation of telomere elongation by enzymatically active telomerase RNPs. Importantly, our findings define an entirely novel, non-coding RNA function for *HIST1H1C*, thus providing a glimpse into a vast space of unknown functions for coding RNAs (*Tay et al., 2014*). Telomere length, as a major determinant of cellular lifespan, is under strict negative regulatory control by various protein factors limiting telomerase action or trimming excessively long telomeres (*Chen et al., 2012*; *Li et al., 2017*). Our findings identify an additional, RNA-based regulatory mechanism serving as a potential proliferative barrier to prevent unlimited cell divisions. While telomerase is repressed in most human somatic cells, its reactivation is a hallmark of malignant transformation, occurring in the majority (~80 – 90%) of cancers (*Kim et al., 1994*). By limiting the available pool of telomerase RNPs via sequestering hTR and hTR-containing complexes, the *HIST1H1C* RNA-hTR interaction might constitute a tumor suppressor mechanism to curb telomere elongation even after upregulation of TERT expression, and thus could contribute to the prevention of malignant transformation, for example in the adult stem cell compartment. Alternative consequences for the interaction in other telomerase-positive human cells, such as in embryonic stem cells or activated lymphocytes, might also warrant further investigation.

Modulating telomerase activity and/or telomere elongation is an attractive and actively pursued target for cancer therapeutics (*Arndt and MacKenzie, 2016*). On the other hand, mutations in genes involved in telomere maintenance result in a list of serious conditions, collectively referred to as

telomeropathies (*Holohan et al., 2014*). As both aberrant telomere lengthening/maintenance and aberrant telomere shortening are of considerable clinical importance, modulating the *HIST1H1C*-hTR RNA-RNA interaction – either by inhibiting or mimicking/facilitating duplex formation – therefore might have significant therapeutic benefits.

# Materials and methods

**Key resources table**

| Reagent type (species) or resource | Designation | Source or reference | Identifiers or link | Additional information |
|---|---|---|---|---|
| Genetic reagent | Lipofectamine 2000 | Thermo Fisher Scientific | Cat#11668019 | |
| Cell line (*Homo sapiens*) | WI38 VA13-2RA | Daniela Rhodes lab (Nanyang Technological University) | N/A | |
| Cell line (*Homo sapiens*) | HT1080 | Peter Dröge lab (Nanyang Technological University) | N/A | |
| Cell line (*Homo sapiens*) | HeLa | Peter Dröge lab (Nanyang Technological University) | N/A | |
| Antibody | Anti-Histone H1.2 antibody [EPR12691] | Abcam | Cat#ab181977 | WB: 1:2000 |
| Antibody | Monoclonal anti-FLAG M2 antibody | Sigma-Aldrich | Cat#F1804; RRID:AB_262044 | WB: 1:1500 |
| Antibody | Anti-p53 (DO-1) antibody | Santa Cruz Biotechnology | Cat#sc126; RRID: AB_628082 | WB: 1:1000 |
| Antibody | Mouse Monoclonal Anti-beta-Actin | Sigma-Aldrich | Cat# A2228, RRID:AB_476697 | WB: 1:5000 |
| Antibody | Goat Polyclonal Anti-Mouse | Dako | Cat# P0447, RRID:AB_2617137 | WB: 1:10000 |
| Antibody | Goat Polyclonal Anti-Rabbit | Dako | Cat# P0448, RRID:AB_2617138 | WB: 1:10000 |
| Recombinant DNA reagent | pU1-hTR(451) | This paper | N/A | |
| Recombinant DNA reagent | pBS-U1-hTR | *Cristofari and Lingner (2006)* | N/A | |
| Recombinant DNA reagent | pBS-U1-hTR-puro | This paper | N/A | |
| Recombinant DNA reagent | pPGK-puro-3x FLAG-wtHIST1H1C | This paper | N/A | |
| Recombinant DNA reagent | pPGK-puro-3x FLAG-silentHIST1H1C | This paper | N/A | |
| Sequence-based reagent | NEBNext Multiplex Oligos for Illumina (Index Primers Set 1) | New England Biolabs | Cat#E7335S | |
| Commercial assay or kit | TeloTAGGG Telomere Length Assay kit | Roche | Cat#12209136001 | |
| Commercial assay or kit | DNeasy Blood and Tissue Kit | Qiagen | Cat#69506 | |
| Commercial assay or kit | Senescence β-Galactosidase Staining Kit | Cell Signaling Technology | Cat#9860 | |
| Commercial assay or kit | QuantiNova SYBR Green RT-PCR Kit | Qiagen | Cat#208154 | |

*Continued on next page*

*Continued*

| Reagent type (species) or resource | Designation | Source or reference | Identifiers or link | Additional information |
|---|---|---|---|---|
| Commercial assay or kit | ThermoScript RT-PCR System | Thermo Fisher Scientific | Cat#11146–024 | |
| Commercial assay or kit | Mycoplasma PCR Detection Kit | Applied Biological Materials | Cat#G238 | |
| Commercial assay or kit | Qubit dsDNA HS Assay Kit | Thermo Fisher Scientific | Cat#Q32851 | |
| Chemical compound, drug | Etoposide | Sigma-Aldrich | Cat#E1383 | |
| Chemical compound, drug | Pierce 16% Formaldehyde (w/v), Methanol-free | Thermo Fisher Scientific | Cat#28906 | |
| Chemical compound, drug | Bovine albumin fraction V | Thermo Fisher Scientific | Cat#15260037 | |
| Chemical compound, drug | Trizol reagent | Thermo Fisher Scientific | Cat#15596026 | |
| Chemical compound, drug | Dynabeads MyOne Streptavidin C1 | Thermo Fisher Scientific | Cat#65001 | |
| Chemical compound, drug | Dynabeads MyOne Silane | Thermo Fisher Scientific | Cat#37002D | |
| Chemical compound, drug | Agencourt AMPure XP | Beckman Coulter | Cat#A63880 | |
| Chemical compound, drug | Ribonuclease H | Thermo Fisher Scientific | Cat#18021014 | |
| Chemical compound, drug | RNase inhibitor, murine | New England Biolabs | Cat#M0314S | |
| Chemical compound, drug | FastAP Thermosensitive Alkaline Phosphatase | Thermo Fisher Scientific | Cat#EF0651 | |
| Chemical compound, drug | T4 polynucleotide kinase | New England Biolabs | Cat#M0201S | |
| Chemical compound, drug | Turbo DNase | Thermo Fisher Scientific | Cat#AM2239 | |
| Chemical compound, drug | Exonuclease I | New England Biolabs | Cat#M0293S | |
| Chemical compound, drug | RQ1 RNase-free DNase | Promega | Cat#M6101 | |
| Chemical compound, drug | ExoSAP-IT PCR Product Cleanup Reagent | Thermo Fisher Scientific | Cat#78200.200 .UL | |
| Chemical compound, drug | NEBNext Ultra II Q5 Master Mix | New England Biolabs | Cat#M0544S | |
| Chemical compound, drug | SYBR Green PCR Master Mix | Thermo Fisher Scientific | Cat#4364344 | |
| Chemical compound, drug | FailSafe enzyme mix | Lucigen | Cat# FSE5101K | |
| Software, algorithm | HISAT2 v2.0.3 | *Kim et al., 2015* | https://ccb.jhu.edu/software/hisat2/index.shtml RRID:SCR_015530 | |
| Software, algorithm | BamTools | *Barnett et al., 2011* | https://github.com/pezmaster31/bamtools RRID:SCR_015987 | |

*Continued on next page*

*Continued*

| Reagent type (species) or resource | Designation | Source or reference | Identifiers or link | Additional information |
| --- | --- | --- | --- | --- |
| Software, algorithm | JAMM peak finder v1.0.7.5 | *Ibrahim et al., 2015* | https://github.com/ mahmoudibrahim/JAMM | |
| Software, algorithm | RNAup | *Gruber et al., 2008* | http://rna.tbi.un ivie.ac.at/cgi-bin/RNAWe bSuite/RNAup.cgi | |
| Software, algorithm | Circos | *Krzywinski et al., 2009* | http://circos.ca/ RRID:SCR_011798 | |
| Software, algorithm | Heatmapper | *Babicki et al., 2016* | http://heatmapper.ca/ | |
| Software, algorithm | DAVID 6.7 | *Huang et al., 2009a, Huang et al., 2009b* | https://david-d.ncifcrf.gov/ RRID:SCR_001881 | |
| Software, algorithm | TeSLA-QUANT | *Lai et al., 2017* | Available as supplementary item in *Lai et al., 2017* | |
| Other | RNA-sequencing data | This paper | SRP123633 https://trace.ncbi.nlm.nih.gov /Traces/sra/?study=SRP123633 | |

## Oligonucleotides

Biotinylated ssODNs used for RNA pull-downs were purchased from Sigma-Aldrich. All other oligo-nucleotides were from Integrated DNA Technologies (IDT).

## Plasmids

U1 promoter sequence and the 451-nt long human telomerase RNA region were amplified from HT1080 genomic DNA and assembled into the pU1-hTR(451) plasmid. Initial RAP-RNA[FA] experiments in VA13 cells were carried out using transient pU1-hTR(451) transfections, achieving hTR expression levels comparable to endogenous amounts in HeLa cells [we estimated hTR expression between ~700 and ~1800 molecules/cell, based on qRT-PCR measurements relative to HeLa cells [~1150 hTR molecules/cell (*Xi and Cech, 2014*)]. The pBS-U1-hTR plasmid (*Cristofari and Lingner, 2006*) contains the U1-hTR cassette and additional downstream sequences from the TERC locus, and can yield ~4–6 fold higher hTR expression levels than pU1-hTR(451). All experiments involving hTR variants were carried out using the pBS-U1-hTR plasmid and its derivatives. Mutations were introduced in pBS-U1-hTR using standard overlap extension PCR protocols. Deleting the 451-nt hTR RNA region resulted in the pBS-U1 pr plasmid, while deleting the terminal stem-loop of the P6b region gave rise to pBS-U1-hTR-ΔP6b. Substitutions of the P6b stem-loop (pBS-U1-hTR-SW, pBS-U1-hTR-RS, and pBS-U1-hTR-P6b_sil) are shown in *Figure 3D* and in *Figure 5—figure supplement 1*. The m1 hTR variant (G414C mutation) was originally described in *Jády et al. (2004)*. For stable transfections, a puromycin expression cassette was inserted in the EcoRV site of pBS-U1-hTR and its variants, resulting in pBS-U1-hTR-puro plasmids.

The HIST1H1C genome region, containing 735 nucleotides upstream of the HIST1H1C start codon, the HIST1H1C ORF, and 452 nucleotides downstream of the stop codon was amplified from HT1080 genomic DNA and inserted into the pPGK-puro plasmid. For control transfections, the HIST1H1C ORF was deleted (mock_HIST1H1C). The 3xFLAG sequence was inserted in-frame at the start codon of HIST1H1C, resulting in pPGK-puro-3xFLAG-wtHIST1H1C. Silent mutations in the TRIAGE sequence were introduced into this plasmid to generate pPGK-puro-3xFLAG-silentHIST1H1C (*Figure 5A*). In order to uncouple RNA and protein expression, a two-nucleotide insertion was added immediately after the 3xFLAG sequence, introducing a frameshift with several stop codons in the HIST1H1C ORF (pPGK-puro-3xFLAG-FS_HIST1H1C and pPGK-puro-3xFLAG-FS_silentHIST1H1C plasmids; *Figure 7A*).

All plasmid constructs were verified by sequencing.

## Cell culture and transfections

Telomerase-negative WI38 VA13-2RA ALT cells were cultured in minimum essential medium (MEM) supplemented with 1 mM sodium pyruvate, 2 mM L-glutamine, 1x MEM NEAA, 10% fetal calf serum, 100 U/ml penicillin, and 100 µg/ml streptomycin. HT1080 and HeLa cells were cultured in Dulbecco's modified Eagle's medium (DMEM) supplemented with 2 mM L-glutamine, 10% fetal calf serum, 100 U/ml penicillin, and 100 µg/ml streptomycin. All cell culture reagents were purchased from Thermo Fisher Scientific. All cell lines used in this study were authenticated by short tandem repeat (STR) genotyping (1st BASE Human Cell Line Authentication Service, Singapore). The absence of mycoplasma contamination was routinely verified by the Mycoplasma PCR detection kit (Applied Biological Materials).

Transfections were carried out using Lipofectamine 2000 reagent (Thermo Fisher Scientific), according to the manufacturer's instructions. For the establishment of stably transfected polyclonal or clonal HT1080 cell lines, transfections were carried out using linearized plasmids, and selection with 3 µg/ml puromycin dihydrochloride (Thermo Fisher Scientific) was initiated 48 hr after transfection.

## RNA antisense purification with formaldehyde cross-linking (RAP-RNA[FA])

We performed RAP-RNA[FA] essentially as described in (*Engreitz et al., 2014*) and in the detailed protocol available at http://www.lncrna-test.caltech.edu/protocols/RAP_Complete_Protocol.pdf, with minor modifications.

### Formaldehyde cross-linking and cell lysis

100 million VA13 or HeLa cells were rinsed in phosphate buffered saline (PBS) and cross-linked with 2% methanol-free formaldehyde (Thermo Fisher Scientific) at 37°C for 10 min, followed by quenching with 500 mM final concentration of glycine at 37°C for 5 min. Cells were rinsed three times with ice-cold PBS and collected in ice-cold scraping buffer (1x PBS, 0.5% bovine albumin fraction V). Cell pellets were collected by centrifugation (1000 g at 4°C for 5 min), and washed again in ice-cold scraping buffer. Aliquots of cell pellets were flash frozen in liquid nitrogen and stored at −80°C until further use.

Pellets from ~20 million cells were lysed in 1 ml lysis buffer (10 mM HEPES pH 7.5, 20 mM KCl, 1.5 mM MgCl$_2$, 0.5 mM EDTA, 1 mM TCEP [tris(2-carboxyethyl)phosphine], 0.5 mM PMSF (phenylmethylsulfonyl fluoride), and 0.1% NP-40) on ice for 10 min, followed by 20 dounces in a 2 ml glass dounce homogenizer. Nuclei were enriched by centrifugation at 4°C, 3300 g for 7 min, and resuspended in 1 ml GuSCN hybridization buffer (20 mM Tris-Cl pH 7.5, 7 mM EDTA, 3 mM EGTA, 150 mM LiCl, 1% NP-40, 0.2% N-lauroylsarcosine, 0.1% sodium deoxycholate, 3M guanidine thyocianate, and 2.5 mM TCEP). RNAs were fragmented using the Bioruptor sonication system (Diagenode) at high-intensity setting for a total of 10 min (30 s ON – 30 s OFF) for VA13 cells and 15 min for HeLa cells, resulting in RNA fragments of ~150–300 nucleotides. Lysates were cleared by centrifugation at 4°C, 16,000 g for 10 min.

### RNA antisense purification

Equimolar mixes of three biotinylated ssDNA ODNs (Sigma-Aldrich) were used for both hTR [hTR_pd1 – 5'-[Biotin]TCGCCCCCGAGAGACCCGCGGCTGACAGAGCCCAACTCTTCGCGGTGGCA (complementary to position 291–340 of hTR); hTR_pd2 – 5'-[Biotin]CCCCGGGAGGGGCGAACGGGCCAGCAGCTGACATTTTTTGTTTGCTCTAG (complementary to position 160–209 of hTR); and hTR_pd3 – 5'-[Biotin]GCGAGAAAAACAGCGCGCGGGGAGCAAAAGCACGGCGCCTACGCCCTTCT (complementary to position 59–108 of hTR)] and U2 snRNA [U2_pd1 – 5'-[Biotin]GGGTGCACCGTTCCTGGAGGTACTGCAATACCAGGTCGATGCGTGGAGTG (complementary to position 139–188 of human U2 snRNA); U2_pd2 – 5'-[Biotin]GACGGAGCAAGCTCCTATTCCATCTCCCTGCTCCAAAAATCCATTTAATA (complementary to position 89–138 of human U2 snRNA); and U2_pd3 – 5'-[Biotin]TATTGTCCTCGGATAGAGGACGTATCAGATATTAAACTGATAAGAACAGA (complementary to position 39–88 of human U2 snRNA)] pull-downs.

To diminish non-specific background, lysates from ~5 million cells were first pre-cleared with 100 µl of Dynabeads MyOne Streptavidin C1 beads (Thermo Fisher Scientific). 1% of the pre-cleared

lysate was set aside for RNA sequencing or qRT-PCR as input RNA sample. 100 pmol of ssDNA ODN mix was added to the pre-cleared lysate and incubated at 37°C for 3 hr with constant shaking at 1200 rpm. 350 µl Dynabeads MyOne Streptavidin C1 beads were washed and resuspended in ¼ bead volume GuSCN hybridization buffer, added to the lysate-ssDNA probe mix, and incubated for a further 30 min at 37°C. Subsequently, beads were washed six times in 1x bead volume GuSCN wash buffer (20 mM Tris-Cl pH 7.5, 10 mM EDTA, 1% NP-40, 0.2% N-lauroylsarcosine, 0.1% sodium deoxycholate, 3M guanidine thyocianate, and 2.5 mM TCEP), with 5 min incubations at 45°C between washes. Finally, beads were washed twice in RNase H elution buffer (50 mM Tris-Cl pH 7.5, 75 mM NaCl, 3 mM MgCl$_2$, 0.125% N-lauroylsarcosine, 0.025% sodium deoxycholate, and 2.5 mM TCEP), and bound RNAs were eluted by incubation at 37°C for 30 min in RNase H elution buffer containing 10U RNase H (Thermo Fisher Scientific, cat no. 18021–014).

## Reversal of cross-links and RNA purification
RNA pull-down and input samples were incubated for 1 hr at 65°C in NLS digestion buffer (20 mM Tris-Cl pH 7.5, 10 mM EDTA, 2% N-lauroylsarcosine, 2.5 mM TCEP, 500 mM NaCl, and 250 µg proteinase K) to remove proteins and reverse the formaldehyde cross-links, followed by nucleic acid purification using Dynabeads MyOne Silane beads (Thermo Fisher Scientific). DNA was digested at 37°C for 30 min in a buffer containing 10 mM Tris-Cl pH 7.5, 1 mM MgCl$_2$, 120 µM CaCl$_2$, 10 mM KCl, 1 mM DTT, 0.002% Triton X-100, 40U murine RNase inhibitor (New England Biolabs), 3U FastAP Thermosensitive Alkaline Phosphatase (Thermo Fisher Scientific), 30U T4 polynucleotide kinase (New England Biolabs), 2U Turbo DNase (Thermo Fisher Scientific), and 20U Exonuclease I (New England Biolabs), followed by RNA purification using Dynabeads MyOne Silane beads.

## RNA sequencing library preparation
20 pmol RiL-19 RNA adapter (5′-[phosph]rArGrArUrCrGrGrArArGrArGrCrGrUrCrGrUrG[3ddC]) (*Engreitz et al., 2014*) was ligated to the purified RNAs by incubation in ligation mix [1x T4 RNA ligase reaction 1 buffer, 9% DMSO, 1 mM ATP, 20% PEG8000 (New England Biolabs), 12U murine RNase inhibitor, and 40U T4 RNA ligase 1 (New England Biolabs)] for 1.5 hr at room temperature, followed by RNA purification using Dynabeads MyOne Silane beads. Reverse transcription was carried out using the ThermoScript RT-PCR system (Thermo Fisher Scientific) and AR17 primer (5′-ACACGACGCTCTTCCGA) (*Engreitz et al., 2014*), according to the manufacturer's instructions. Excess RT primers were removed by the ExoSAP-IT PCR product cleanup reagent (Thermo Fisher Scientific). Remaining ssDNA probes were removed by incubating the samples with Dynabeads MyOne Streptavidin C1 beads at 60°C for 15 min with shaking at 1200 rpm in C1 binding buffer (10 mM Tris-Cl pH 7.5, 250 mM LiCl, 20 mM EDTA, 0.1% Triton X-100). Before the second adapter ligation, RNAs were digested by incubation in 100 mM NaOH at 70°C for 10 min, followed by neutralization by acetic acid and cDNA purification using Dynabeads MyOne Silane beads. 40 pmol 3Tr3 DNA adapter (5′-[phosph]AGATCGGAAGAGCACACGTCTG[3ddC]) (*Engreitz et al., 2014*) was ligated to the purified cDNAs by incubation in ligation mix (1x T4 RNA ligase reaction 1 buffer, 4% DMSO, 1 mM ATP, 24% PEG8000, and 50U T4 RNA ligase 1) overnight at room temperature, followed by nucleic acid purification using Dynabeads MyOne Silane beads. Samples were enriched by PCR using NEBNext Ultra II Q5 master mix (New England Biolabs) with NEBNext Multiplex Oligos for Illumina (Index Primers Set 1) (New England Biolabs). PCR conditions were as follows: 98°C for 30 s, 4 cycles of (98°C for 10 s – 67°C for 30 s – 72°C for 30 s), and a variable number of cycles of (98°C for 10 s – 72°C for 30 s). Input samples required 4–6 cycles, pull-down samples 10–12 cycles, and negative controls 14–16 cycles at this stage to obtain sufficient DNA quantities for sequencing. PCR products were purified by Agencourt AMPure XP beads (Beckman Coulter), quantified using the Qubit dsDNA HS Assay kit (Thermo Fisher Scientific), and submitted for Illumina sequencing.

## RNA-sequencing and bioinformatic analysis
Illumina HiSeq RNA-sequencing on multiplexed samples was carried out to obtain ~20 million 100-nt paired-end reads per sample. Adaptor sequences and low quality bases were trimmed from raw sequencing reads using cutadapt (*Martin, 2011*). Reads were aligned to the human genome (GRCh38) using HISAT2 (v2.0.3) (*Kim et al., 2015*) with default parameter settings. Bam files were filtered to remove PCR duplicates, reads derived from transcription of the hTR expression plasmid,

mitochondrial transcripts, and ribosomal RNA sequences (*Engreitz et al., 2014*). Peaks enriched in the pull-down over input samples were identified using the JAMM universal peak finder (v1.0.7.5) (*Ibrahim et al., 2015*) with the following parameters: *-m narrow -r peak*. To increase the stringency of our analysis, peaks were further filtered by requiring a minimum peak score of 50,000 (obtained using the JAMM software), a minimum 4-fold enrichment (defined as normalized reads in the pull-down sample/normalized reads in the input sample) over the peak region, and a minimum number of 20 reads supporting the peak. Regions enriched in any of the control pull-downs from hTR-negative, mock-transfected VA13 cells were excluded from further analysis. Peaks passing these filtering criteria, together with the unfiltered output files, are listed in *Supplementary file 1*. Enriched transcripts were ranked based on peak scores obtained with the JAMM software (*Ibrahim et al., 2015*). Putative hTR-target RNA interactions were predicted with the RNAup software (*Gruber et al., 2008*). Functional annotation analysis was carried out using DAVID 6.7 (*Huang et al., 2009a*; *Huang et al., 2009b*), using all human genes as background.

## qRT-PCR

For qRT-PCRs verifying the enrichment of selected transcripts upon hTR pull-down, RNAs were purified as described in the RAP-RNA[FA] section. For measuring cellular RNA expression levels, total cellular RNA was extracted using Trizol reagent (Thermo Fisher Scientific) and treated by RQ1 DNase (Promega). qRT-PCR reactions were carried out using QuantiNova SYBR Green RT-PCR kit (Qiagen), according to the manufacturer's protocol. Amplification conditions were as described in *Zhang et al. (2015)*, followed by melting curve analysis using the Bio-Rad CFX96 qPCR instrument. Serial dilutions of positive samples were included on each plate for each target RNA to obtain standard curves for relative quantification.

qRT-PCR primer sequences are listed in *Table 1*.

## Telomere restriction fragment analysis (TRF)

Telomere restriction fragments were analyzed using the TeloTAGGG Telomere Length Assay kit (Roche). Briefly, cells were harvested by trypsinization, washed in PBS and collected by centrifugation at 400 g for 4 min. Genomic DNA was isolated using DNeasy Blood and Tissue Kit (Qiagen), digested with HinfI and RsaI restriction enzymes (New England Biolabs) and separated by gel electrophoresis either on 0.8% agarose gels at 50V overnight in 1X TBE buffer or (to resolve elongated telomeres at later time points) on 1% megabase agarose gels (Bio-Rad) using a CHEF DRII equipment (Bio-Rad) under the following conditions: 120° field angle, 5 to 30 s switch times, 5 V/cm and 14°C for 14 hr in 1X TAE. Following the resolution of DNA fragments, DNA was transferred to a positively charged nylon membrane (Roche) by Southern blotting and hybridized with a digoxigenin-labelled telomeric probe. Membranes were exposed to X-ray film (Carestream) and developed in X-OMAT 2000 Processor (Kodak). Mean telomere lengths were calculated as described in *Kimura et al. (2010)*.

## Telomere shortest length assay (TeSLA)

The TeSLA procedure was carried out as described in *Lai et al. (2017)*, with minor modifications. Oligonucleotide sequences for the ligation and amplification reactions were published in *Lai et al. (2017)*.

Briefly, 50 ng of genomic DNA was ligated with TeSLA-T oligonucleotides and then digested with CviAII, BfaI, NdeI, and MseI restriction enzymes (New England Biolabs), followed by shrimp alkaline phosphatase (New England Biolabs) treatment. The digested DNA was ligated with double-stranded TeSLA adapters, and 30 pg of the ligated DNA was subsequently used for long-range PCR amplifications. Amplification reactions were carried out in 25 µl volume, using 2.5 units of FailSafe Enzyme Mix (Lucigen) with FailSafe buffer H and 0.25 µM primers (AP and TeSLA-TP). After the initial melt at 94°C for 2 min, 25 PCR cycles were carried out (94°C for 15 s, 60°C for 30 s, and 72°C for 15 min). Amplified PCR products were resolved on a 1.2% agarose gel at 50V overnight in 1X TBE buffer. Southern blotting and telomere signal detection was performed using the TeloTAGGG Telomere Length Assay kit (Roche), as described for the TRF assay. The TeSLA-QUANT software was used for image quantification and statistical analysis, as described in *Lai et al. (2017)*.

**Table 1.** qRT-PCR primers used in this study.

| Primer name | Sequence (5' to 3') | Reference |
|---|---|---|
| hTR_fw | GAAGAGGAACGGAGCGAGTC | *Xi and Cech (2014)* |
| hTR_rev | ATGTGTGAGCCGAGTCCTG | *Xi and Cech (2014)* |
| GAPDH_fw | ACAGCAACAGGGTGGTGGAC | *Xi and Cech (2014)* |
| GAPDH_rev | GACCATTGCTGGGGCTGGTG | *Xi and Cech (2014)* |
| 28S_rRNA_fw | GTGTCAGAAAAGTTACCACA | *Rocchi et al. (2014)* |
| 28S_rRNA_rev | ATTATGCTGAGTGATATCCC | *Rocchi et al. (2014)* |
| HIST1H1C(272-386)_fw | GCACTCTGGTGCAAACGAAAG | |
| HIST1H1C(272-386)_rev | TTAGGTTTGGTTCCGCCCG | |
| HIST1H1C(53-145)_fw | CCCCTGTAAAGAAGAAGGCGG | |
| HIST1H1C(53-145)_rev | CCACAGCCTTGGTGATGAGC | |
| HIST1H4D_fw | CCAAGCGTCACCGTAAGGTAT | |
| HIST1H4D_rev | AAACTTTCAGCACTCCGCGA | |
| HIST1H3B_fw | AGACAGCTCGGAAATCCACC | |
| HIST1H3B_rev | AACGGTGAGGCTTTTTCACG | |
| scaRNA2_fw | TTTAGGGAGGGAGAGCGGC | |
| scaRNA2_rev | CTCACGCGTCCACTCACAC | |
| scaRNA11_fw | GTCCTGGCCTATTTTTCTGCTCC | |
| scaRNA11_rev | CGGCCCTCCTAAACTGAGAGG | |
| scaRNA12_fw | TGGGGACTACAGATGAGATCTGA | |
| scaRNA12_rev | GATCCAAGGTTGCGCTCAGG | |
| TPT1_fw | GGCCTTTTCCGCCCGC | |
| TPT1_rev | CTAGCTTAGCACGAGCCTGA | |
| FLNA_fw | CGGTGATCACTGTGGACACTA | |
| FLNA_rev | ATTCTCCACCACGTCCACATC | |
| IFITM3_fw | GAGCTCTGCCCATGACCTG | |
| IFITM3_rev | GCTGATACAGGACTCGGCTC | |

DOI: https://doi.org/10.7554/eLife.40037.022

## Real-time quantitative telomeric repeat amplification protocol (RQ-TRAP)

Cells were trypsinized, washed once in ice-cold PBS, and resuspended at $10^6$ cells/ml in CHAPS lysis buffer containing 400 mM NaCl (10 mM Tris-Cl pH 7.5, 400 mM NaCl, 1 mM $MgCl_2$, 1 mM EGTA, 0.5% 3-[(3-cholamidopropyl)dimethylammonio]−1-propanesulfonate (CHAPS), 10% glycerol, 1x complete protease inhibitor cocktail without EDTA (Roche), 5 mM β-mercaptoethanol), as described in *Cristofari et al. (2007)*. Following incubation for 30 min on ice, cell debris was removed by centrifugation at 4°C for 10 min at 12,000 g. The supernatant was aliquoted, flash frozen in liquid nitrogen and stored at −80°C until further use.

Protein concentration of the extracts was determined using the Bradford assay (Bio-Rad). Amplification reactions were carried out in 20 µl volume, using SYBR Green Universal Master Mix (Thermo Fisher Scientific), 1 mM EGTA, 80 ng TS primer (5'- AATCCGTCGAGCAGAGTT) (*Herbert et al., 2006*), 80 ng ACX primer (5'- GCGCGGCTTACCCTTACCCTTACCCTAACC) (*Herbert et al., 2006*), and 1 µl cell lysate (diluted to between 200 ng/µl and 0.3 ng/µl protein concentration).

Using the Bio-Rad CFX96 qPCR instrument, samples were incubated for 30 min at 30°C, for 10 min at 95°C, and amplified in 40 PCR cycles with 15 s at 95°C and 60 s at 60°C. Relative telomerase activities were determined using a standard curve from 5-fold serial dilutions of wild-type hTR-transfected HT1080 cells. Samples were also serially diluted to verify that the measurements were carried out in the linear range of the assay. qTRAP measurements were carried out in triplicates and

repeated at least twice for each sample. Heat-inactivated wild-type hTR-transfected HT1080 cell lysates and CHAPS buffer were used as negative controls, verifying that the amplification products were due to telomerase activity.

## Western blotting

Cells were lysed by sonication in ice-cold RIPA buffer containing 1x Protease Inhibitor Cocktail (Roche). Proteins were then separated by SDS-PAGE and transferred to 0.2 µm PVDF membranes (Bio-Rad). Membranes were blocked with Superblock T20 (TBS) blocking buffer (Thermo Fisher Scientific) and probed with primary [anti-Histone H1.2 (Abcam, ab181977, 1:2000 dilution); anti-FLAG M2 (Sigma, F1804, 1:1500 dilution), or anti-beta-actin (Sigma, A2228, 1:5000 dilution)] and secondary [(polyclonal goat anti-rabbit (Dako, P0448, 1:10,000 dilution) or polyclonal goat anti-mouse (Dako, P0447, 1:10,000 dilution)] antibodies or HRP-conjugated p53 antibody (DO1 – sc126, Santa Cruz Biotechnology, 1:1000 dilution). Reactive bands were visualized with Immobilon western chemiluminescent HRP substrate (Millipore) in a luminescence imager (LAS4000, Fujifilm).

## Senescence-associated-β-galactosidase staining

Clonal HT1080 cell lines stably expressing exogenous wtHIST1H1C were seeded in 6-well plates and cultured for 4–5 days prior to staining. Cells were fixed and stained according to the manufacturer's protocol (Senescence β-Galactosidase Staining Kit, Cell Signaling #9860). Stained cells were imaged with an inverted microscope (Olympus, CKX41) with PixeLINK colour megapixel firewire camera (PL-A662). The percentage of SA-β-gal positive cells were determined by manually counting the positive cells across four randomly selected microscope frames and then normalized to the total number of cells. Staining was performed in triplicates between 8 (1 sample) and 10 weeks (two samples) post-transfection, and cell counts were pooled for the analysis. HT1080 cells treated with 10 µM of the topoisomerase type 2 inhibitor Etoposide (Sigma) for 24 hr followed by 24 hr recovery served as positive control. Untransfected HT1080 cells served as negative control.

## Statistical analyses

Statistical analyses were performed as described in the figure legend for each experiment. All data are presented as mean ± s.d., unless otherwise noted in the legends. All data shown are representative of two or more independent experiments, unless otherwise indicated.

### Data availability

RNA-seq data that support the findings of this study have been deposited in the NCBI Sequence Read Archive (SRA) under the accession code SRP123633.

## Acknowledgements

The VA13 cell line was kindly provided by D Rhodes (Nanyang Technological University, Singapore), and the pBS-U1-hTR plasmid by J Lingner (ISREC, Lausanne, Switzerland). We are grateful to D Rhodes for comments on the manuscript, D Lane and F Ghadessy for the conjugated p53 monoclonal antibody, and D Barta for assistance with figure preparation. This work was supported by the Ministry of Education of Singapore (MOE2012-T3-1-001 grant to PD). OD and PFO were supported by the Singapore Biomedical Research Council and the Singapore Agency for Science, Technology and Research (A*STAR).

## Additional information

### Funding

| Funder | Grant reference number | Author |
| --- | --- | --- |
| Ministry of Education - Singapore | MOE2012-T3-1-001 | Peter Dröge |
| Singapore Biomedical Research Council | | Oliver Dreesen |
| Agency for Science, Technology and Research | | Oliver Dreesen |

The funders had no role in study design, data collection and interpretation, or the decision to submit the work for publication.

### Author contributions
Roland Ivanyi-Nagy, Conceptualization, Formal analysis, Investigation, Visualization, Methodology, Writing—original draft; Syed Moiz Ahmed, Formal analysis, Investigation, Visualization; Sabrina Peter, Priya Dharshana Ramani, Peh Fern Ong, Investigation; Oliver Dreesen, Resources, Methodology, Writing—review and editing; Peter Dröge, Conceptualization, Supervision, Funding acquisition, Methodology, Writing—original draft, Project administration

### Author ORCIDs
Roland Ivanyi-Nagy ⓘ http://orcid.org/0000-0001-7275-327X
Syed Moiz Ahmed ⓘ http://orcid.org/0000-0002-6709-0245
Oliver Dreesen ⓘ http://orcid.org/0000-0003-1148-3557
Peter Dröge ⓘ http://orcid.org/0000-0001-5447-738X

### Decision letter and Author response
Decision letter https://doi.org/10.7554/eLife.40037.028
Author response https://doi.org/10.7554/eLife.40037.029

## Additional files

### Supplementary files
• Supplementary file 1. Enriched peaks identified in hTR pull-down samples, using the JAMM universal peak finder (*Ibrahim et al., 2015*).
DOI: https://doi.org/10.7554/eLife.40037.023

• Transparent reporting form
DOI: https://doi.org/10.7554/eLife.40037.024

### Data availability
Sequencing data have been deposited in the NCBI Sequence Read Archive (SRA) under the accession code SRP123633 (SRR6255719-SRR6255732).

The following dataset was generated:

| Author(s) | Year | Dataset title | Dataset URL | Database and Identifier |
|---|---|---|---|---|
| Roland Ivanyi-Nagy, Syed Moiz Ahmed, Sabrina Peter, Priya Dharshana Ramani, Peter Dröge | 2018 | Human telomerase RNA-RNA interactome | https://www.ncbi.nlm.nih.gov/sra/?term=SRP123633 | NCBI Sequence Read Archive, SRP123633 |

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
