## [Decision Letter]

Thank you for submitting your article "The RNA interactome of human telomerase RNA reveals a coding-independent role for a histone mRNA in telomere homeostasis" for consideration by *eLife*. Your article has been reviewed by three peer reviewers, one of whom is a member of our Board of Reviewing Editors, and the evaluation has been overseen by James Manley as the Senior Editor. The following individuals involved in review of your submission have agreed to reveal their identity: Thomas R Cech (Reviewer #2); Marlene Oeffinger (Reviewer #3).

The reviewers have discussed the reviews with one another and the Reviewing Editor has drafted this decision to help you prepare a revised submission.

Thanks for submitting your manuscript to *eLife* for potential publication. As you read above, the paper now has been evaluated by three reviewers and I am pleased to report that they all very much support it. Indeed, they agreed that the manuscript reports an exciting and potentially very important analysis of the "RNA interactome" of human telomerase RNA (hTR). They were impressed by the depth and breadth of the analyses that include pull-down assays, mutational tests, overexpression experiments, telomere length measurements, and cellular senescence assays. Furthermore, the reviewers commented on the fact that the experiments are generally well-controlled, repeated a good number of times, and clearly explained.

Nevertheless, there were issues for which the reviewers felt that an amelioration would be great. I will subdivide those into one major issue, for which further experimental evidence is required, and some other issues that you may address experimentally and/or by editorial changes.

Major issue:

All experiments in which mutated RNAs are used and that are aimed at documenting biological effects of a loss of the supposed RNA-RNA interaction were done incorporating mutations in only one of the two partner RNAs. A rather standard experiment for these kinds of approaches is to also mutate both RNAs at the same time in such a way that the stipulated base-pairing interaction is re-established, albeit with the mutated sequences. Such an experiment would go a long way in supporting your claim that the phenotypes you see for the individual mutations are due to the presumed loss of RNA-RNA interaction. You could do this experiment rather easily in the way that you did the individual mutations (Figure 3E for example) to show that the two mutated RNAs do interact and can be pulled down.

As one reviewer put it: "In the last paragraph of the subsection “*HIST1H1C* RNA specifically interacts with hTR”, the authors conclude that sequence complementarity between the TRIAGE region of the histone mRNA and P6b of hTR is necessary for the interaction. However, mutating one partner in the proposed interaction (in this case, P6b of hTR) only shows that these sequences are necessary, it doesn't indicate that they're necessary *because* they base-pair with the other partner. Subsection “Disrupting the *HIST1H1C*-hTR RNA-RNA interaction leads to increased telomere elongation”, and also the title of Figure 4, these words claim a causal relationship between disrupting the interaction and increased telomere length. But again, it's the mutation in hTR that causes both disruption of interaction and increased telomere length, and it's a plausible MODEL that it does so via disruption of an RNA-RNA interaction. (1) "the TRIAGE sequence is responsible for the observed phenotype" is a correct conclusion from the data, but (2) "telomere length is regulated by the.…RNA-RNA interaction" was not shown, because only one partner was mutated (in this case, the TRIAGE sequence in the mRNA). "80 RNA species interacting with hTR" should be qualified to say "80 RNA species interacting directly or indirectly with hTR."

The qualifications requested in the above paragraph would go away if the authors could perform a "killer experiment" to directly test for RNA-RNA base-pairing. This is the classic requirement in the RNA field: mutate sequence 1 and lose activity, mutate sequence 2 and lose activity, make the compensatory double-mutant and regain activity. The authors have set themselves up beautifully to do this experiment: in the subsection “HIST1H1C regulates telomere length as a non-coding RNA”, they express a mutated mRNA "silentHIST1HC" with half of the TRIAGE nucleotides mutated. This mutant does not show the telomere elongation phenotype seen with the WT HIST1H1C mRNA. If they were to simultaneously mutate P6b in hTR so that it matched the sequence of silentHIST1HC, their model would predict that the double mutant would regain telomere elongation and regain pull-down."

---

## [Author Response]

Major issue:[…] As one reviewer put it: "In the last paragraph of the subsection “HIST1H1C RNA specifically interacts with hTR”, the authors conclude that sequence complementarity between the TRIAGE region of the histone mRNA and P6b of hTR is necessary for the interaction. However, mutating one partner in the proposed interaction (in this case, P6b of hTR) only shows that these sequences are necessary, it doesn't indicate that they're necessary because they base-pair with the other partner. Subsection “Disrupting the HIST1H1C-hTR RNA-RNA interaction leads to increased telomere elongation”, and also the title of Figure 4, these words claim a causal relationship between disrupting the interaction and increased telomere length. But again, it's the mutation in hTR that causes both disruption of interaction and increased telomere length, and it's a plausible MODEL that it does so via disruption of an RNA-RNA interaction. (1) "the TRIAGE sequence is responsible for the observed phenotype" is a correct conclusion from the data, but (2) "telomere length is regulated by the.…RNA-RNA interaction" was not shown, because only one partner was mutated (in this case, the TRIAGE sequence in the mRNA). "80 RNA species interacting with hTR" should be qualified to say "80 RNA species interacting directly or indirectly with hTR."The qualifications requested in the above paragraph would go away if the authors could perform a "killer experiment" to directly test for RNA-RNA base-pairing. This is the classic requirement in the RNA field: mutate sequence 1 and lose activity, mutate sequence 2 and lose activity, make the compensatory double-mutant and regain activity. The authors have set themselves up beautifully to do this experiment: in the subsection “HIST1H1C regulates telomere length as a non-coding RNA”, they express a mutated mRNA "silentHIST1HC" with half of the TRIAGE nucleotides mutated. This mutant does not show the telomere elongation phenotype seen with the WT HIST1H1C mRNA. If they were to simultaneously mutate P6b in hTR so that it matched the sequence of silentHIST1HC, their model would predict that the double mutant would regain telomere elongation and regain pull-down."

We carried out a conceptually similar experiment, incorporating mutations in the P6b stem-loop of hTR that introduced complementarity to a region of *HIST1H1C* distinct from the TRIAGE sequence (RS mutant in Figures 3 and 4). In that case, complementarity in itself was not sufficient to ‘rescue’ the hTR-*HIST1H1C* association. We speculated that the *HIST1H1C* region complementary to the RS-hTR variant might not be accessible for hybridization. Alternatively, P6b-TRIAGE duplex formation might be facilitated by an RNA-binding protein with some sequence-specificity/preference.

Now, based on the reviewers’ recommendations, we have introduced mutations in the P6b stem-loop of hTR that re-establish complementarity to the silentHIST1H1C mRNA. As shown in a new Figure 5—figure supplement 1, this hTR variant specifically interacts with and pulls down the silentHIST1H1C mutant mRNA, but not wtHIST1H1C. Similarly, wild-type hTR only pulled down the wild-type HIST1H1C mRNA (either endogenous or exogenous), but not the silentHIST1H1C variant. These results clearly demonstrate a direct RNA-RNA interaction between hTR and *HIST1H1C*, and confirm the requirements for the TRIAGE and P6b regions for duplex formation. Unfortunately, the mutant hTR does not support the assembly of catalytically active telomerase RNPs, possibly due to the disruption of the P6b stem-loop structure, which might also interfere with the correct folding of the adjacent P6.1 region.